## Registered report

psychology

free will, beliefs, attitudes, experimental philosophy, open science

# Relating free will beliefs and attitudes

David Wisniewski[1], Emiel Cracco[2],
Carlos González-García[1,3] and Marcel Brass[1,4]

[1]Department of Experimental Psychology, Ghent University, Henri Dunantlaan 2, 9000 Ghent, Belgium
[2]Department of Experimental Clinical and Health Psychology, Ghent University, Ghent, Belgium
[3]Mind, Brain and Behavior Research Center, Department of Experimental Psychology, University of Granada, Granada, Spain
[4]Berlin School of Mind and Brain/Department of Psychology/Excellence Cluster 'Science of Intelligence', Humboldt Universität zu Berlin, Berlin, Germany

 DW, 0000-0003-4793-1171; EC, 0000-0003-4043-5992;
CG-G, 0000-0001-6627-5777; MB, 0000-0002-3364-4019

Most people believe in free will, which is foundational for our sense of agency and responsibility. Past research demonstrated that such beliefs are dynamic, and can be manipulated experimentally. Much less is known about free will attitudes (FWAs; do you value free will?), whether they are equally dynamic, and about their relation to free will beliefs (FWBs). If FWAs were strongly positive, people might be reluctant to revise their beliefs even in the face of strong evidence to do so. In this registered report, we developed a novel measure of FWAs and directly related FWBs and attitudes for the first time. We found FWBs and attitudes to be positively related, although to a lesser degree than determinism or dualism beliefs/attitudes. Nevertheless, an experimental manipulation technique aimed at reducing FWBs (Crick text) showed remarkably specific effects on FWBs only, and no effects on FWAs. Overall, these results provide valuable new insights into laypeople's views on free will by including a novel measure of FWAs. They also provide evidence for the validity of a common experimental technique that has been rightfully criticized in the literature lately.

## 1. Introduction

Whether free will exists or not has been a hotly debated topic in the philosophic discourse for centuries [1]. More recently, experimental philosophy has started to map out lay beliefs about free will [2–7]. This undertaking has important theoretical implications, as philosophical theories on free will are often based on intuitions, and knowing whether such intuitions are widespread or not can be valuable both for academic research [8] and beyond. Additionally,

**Author for correspondence:**
David Wisniewski
e-mail: david.wisniewski@ugent.be

we know that free will beliefs (FWBs) are dynamic [9,10] and partly dependent on culture [11]. Some fear that because of this dynamic property, FWBs in laypeople might decrease in the future, especially since recent neuroscientific findings apparently demonstrate that our choices are determined by unconscious brain activity [12–14]. This might have potentially negative consequences [15] since our sense of responsibility and even our criminal punishment systems are built on a belief in free will [16]. In order to understand and predict any such changes, we first need to map out laypeople's baseline views. Without understanding those, understanding how they might change will prove difficult. Thus, there is a clear need to better understand lay concepts of free will, both for theoretical and practical reasons.

## 1.1. Free will beliefs and attitudes

In most of this literature, FWBs are conceptualized as agreement to specific propositional statements (e.g. 'How people's lives unfold is completely up to them', [17]). Agreeing to these statements is taken as evidence for a belief in free will. Understanding beliefs as agreement to propositional statements is important, as FWBs are potentially linked to e.g. social behaviour ([9,18,19]; but see [20,21]). This arguably captures only a part of laypeople's broader views on free will though.

A large literature has shown that next to beliefs, attitudes are equally important and are independently predictive of behaviour [22–24]. Attitudes toward free will (FWAs), which can be measured using valence or liking judgements (e.g. 'I value free will'), are, therefore, potentially related to behaviour just like FWBs are. While evaluations/liking judgements are not valid arguments in academic discourse, this does not necessarily apply to lay theories [11,25]. Laypeople can and do have clear attitudes toward concepts like free will [26], and such attitudes might be used to inform beliefs (I believe in free will because I value having free will). Critically, however, we currently know very little about FWAs, and cannot draw strong conclusions about whether or not this is indeed the case. In fact, we do not even know whether FWBs and FWAs are related. Some initial evidence suggests that FWAs are strongly positive [26], mirroring research on FWBs [11,17,27]. Yet, given that no study so far has acquired both FWBs and FWAs simultaneously, it remains difficult to conclude how FWAs and FWBs are related. Here, we performed such a direct comparison for the first time, and expected them to be positively related (Hypothesis 1).

Most previous studies on FWBs used an experimental FWB manipulation (for a review of the literature see [21]). Participants are assigned to an anti-free will or control group, and are e.g. given free-will-related texts to read [19], are asked to think about free-will-related statements [28], or watch a free-will-related video [29]. These techniques have been shown to reduce FWBs [21], which demonstrates that FWBs are dynamic [9], and we expected them to alter FWBs in this study as well (Hypothesis 2). Although many studies demonstrated such effects, they are often small and not highly consistent across participants and studies [9,21]. Here we argue that understanding how FWBs and FWAs are related might help explain why experimental belief manipulation effects are small and inconsistent.

It seems reasonable to assume that a person with a very positive attitude toward free will be highly reluctant to give up their FWBs while participating in a short psychological experiment [30,31]. By contrast, a person who has a relatively more negative attitude toward free will be more prone to give up their FWBs when confronted with anti-free will messages (Hypothesis 3). However, to properly assess the role of baseline attitudes in belief change, both beliefs and attitudes need to be measured before and after a FWB manipulation. By contrast, almost all past research only measured beliefs after a manipulation, making it impossible to test this hypothesis (e.g. [10], but see [32]). Relatedly, baseline FWBs might have similar effects (Hypothesis 4). Just like with baseline attitudes, a person strongly believing in free will might be more reluctant to give up their beliefs than a person disbelieving in free will, or being agnostic/unsure about its existence.

## 1.2. Specificity of free will belief manipulations

While it is well established that FWBs are dynamic, many past studies (at least implicitly) assume that manipulation effects are specific to free will beliefs. This important assumption has received little attention in the past and has not been directly tested, and here we argue that at least some FWB manipulations also affect free will attitudes, which would make them less specific than we currently think. One of the most common belief manipulation techniques is letting participants read a passage from Francis Crick's book *The Astonishing Hypothesis* [33, pp. 265–268]. In this passage (as used in e.g. [19]), the author clearly argues against free will, based on the notion that conscious choices and reasons are merely *post hoc* explanations of unconscious brain activity. While this should reduce belief

in free will, it does so by describing free will as an essentially unnecessary construct with clearly negative connotations. It suggests that people who believe in free will 'confabulate' an explanation of their own behaviour, and jump 'to the simplest conclusion'. Free will is further compared with flat Earth theory, only reinforcing the negative connotations of free will in this text. Thus, it seems reasonable to assume that this manipulation is not specific to beliefs, but also makes attitudes toward free will more negative (Hypothesis 5).

If this were the case, there would be consequences for both the methodology of free will research and current theories of FWBs. First, directly relating FWBs and attitudes opens up novel avenues for theories of FWBs and belief change. In the attitude domain, there are already strong theories of how and when attitudes influence behaviour, which are currently lacking in the FWB domain [31]. Adapting theories like the elaboration likelihood model [34] might thus give a more precise explanation of how manipulations might change FWBs and attitudes, and whether such changes will in turn affect behaviour or not. Depending on e.g. motivational state or the ability to fully process incoming information, free will manipulations might either lead to temporary attitude shifts that are unpredictive of behaviour, or to enduring attitude shifts that are predictive of behaviour. Current theories of FWBs have so far been unable to offer such predictions. Second, our research informs how future studies measure the effects of FWB manipulations. Past research commonly relies on FWB measures as a manipulation check. The manipulation is assumed to have worked if e.g. FWBs are lower in an anti-free will group, as compared with a control group [10,32]. Conversely, no significant results in such a manipulation check are typically taken as evidence for a failed manipulation [35]. If downstream effects of FWB were to be mainly driven by attitude change, however, claiming a failed manipulation based on a belief manipulation alone would not be valid. Thus, it might be that some 'failed' experiments could be reinterpreted as only failing to affect beliefs specifically, but not attitudes. Bridging the gap between the free will and attitude literatures opens up a new line of research, with ample room for theory development and refinement of research methodology.

## 1.3. The present investigation

In this registered report, we performed a study to directly tackle these key open issues, and bridge the gap between FWB and attitude research. In a first experimental session, participants rated their baseline free will attitudes and beliefs. Attitudes were measured using a novel questionnaire, derived from the free will inventory (FWI) [17]. After about two weeks, they performed a second experimental session, in which they went through a free will belief manipulation (Crick text), after which both attitudes and beliefs were measured again. This design allowed us to address our three main research questions. First, to assess the relation of FWBs and attitudes, we correlated both within the first session. This allowed us to test whether and to which degree beliefs and attitudes are related. Second, to test whether baseline beliefs and attitudes affect FWB manipulations, we used the pre measurements as moderators and tested whether the manipulation strength was modified by them. This allowed us to assess whether stronger baseline beliefs or attitudes decreased the effectiveness of the belief manipulation. Third, to test whether FWB manipulations are specific to beliefs only, we tested whether the manipulation only affected belief measures, or whether it also affected attitude measures.

# 2. Methods

## 2.1. Participants and power analysis

We performed an *a priori* power analysis to determine the required sample size to reliably detect effects of our FWB manipulation ($\alpha = 0.05$ and $\beta = 0.90$). Such manipulations are notoriously small [21], and we thus conservatively assumed only a small effect size ($d = 0.20$). To test whether the manipulation worked, we computed a difference score for each participant (FWBs after the manipulation − FWBs before the manipulation). This difference score is our primary measure of the manipulation strength, and we entered it into a two-sample *t*-test to compare both groups (Anti-free-will or AntiFW, Control; for details on the design and measures see below). In order to reliably detect group differences here, the required sample size was $n = 528$ in each group or $n = 1056$ in total.

All participants were recruited and paid online through the Prolific Academic platform. The inclusion criteria were: age between 18 and 60, first language: English, country of birth (USA), current country of residence (USA), at least 30 prior submissions on the platform, and at least a 90% approval rate.

Restricting the sample to the USA makes this study more comparable to prior large-scale studies on FWBs [11,17].

The final sample comprised $n = 528$ participants in each group. The mean age was 35.85 years (range 18–60 years); 49.53% of the sample was female, 49.43% male and 1.04% identified as diverse. A large majority of the sample had a university degree (4.92% doctoral, 20.36% graduate, 46.40% undergraduate); 25.76% had a high-school degree, 1.52% a secondary degree and 0.09% no formal degree. Regarding faith, 47.73% reported to be Christian, 44.51% reported to be non-religious. All other faiths were rarely reported (3.41% other, 1.89% Judaic, 0.66% Muslim, 0.37% Buddhist, 0.57% Hindu).

## 2.2. Procedure

### 2.2.1. Pre session

The whole experiment was conducted online in a web browser. Upon signing up, participants first gave their informed consent.

**Demographic data**. Participants first provided demographic information. They were asked about their age, gender (male, female, other, prefer to not say) and highest level of education (no formal education, secondary education, high-school, undergraduate degree, graduate degree, doctorate degree, prefer not to say). For exploratory purposes, we also asked about religious affiliations (Christianity—Catholic, Christianity—Protestant, Christianity—Orthodox, Christianity—Other, Islam—Sunni, Islam—Shi'ite, Islam—Other, Judaism, Buddhism—Theravada, Buddhism—Mahayana, Buddhism—Vajrayana, Hinduism, Taoism, non-religious, other, prefer not to say), and the strength of their religious beliefs (7-point scale from 'very weak' to 'very strong').

**Free will beliefs and attitudes**. Subsequently, participants rated their FWAs and FWBs. Before items were presented, participants were shown the following instructions: 'You will now see a number of statements on screen. Please read these statements carefully, think about them and indicate your response using the options on screen. You will always be asked first whether you believe the statement to be true or false. After that, you will be asked whether you like what the statement says or not. Please attend to what is asked, read all response options carefully to avoid errors, and respond according to your true views.' Then, we presented two training items on screen. First participants were asked to respond to 'Humans can walk on the surface of the sun and survive.' (responses: 7-point Likert scale, 1 = strongly disagree, 4 = neither agree nor disagree, 7 = strongly agree), then they were asked to respond to 'If it were up to me, I would like for humans to be able to walk on the surface of the sun and survive.' (responses: 7-point Likert scale, 1 = strongly dislike, 4 = neither like nor dislike, 7 = strongly like). These two items were presented in order to familiarize participants with rating both their beliefs and attitudes toward a similar item. Then, the following instruction was presented: 'Next, you will see a number of similar statements. Please respond to them truthfully, just like you did now.'

We then presented the 15 items of the FWI, intermixed with modified versions of the same items that were optimized to assess attitudes. Specifically, we first presented each FWI item in its original version [17], immediately followed by a modified version rephrased for assessing attitudes instead of beliefs (see electronic supplementary material, Methods 1). For instance, we first presented 'People always have the ability to do otherwise.' (belief version) and then 'I would like if people always had the ability to do otherwise.' (attitude version). The same procedure was applied to all other FWI items. Each belief item was scored on a 7-point Likert scale (1 = strongly disagree, 4 = neither agree nor disagree, 7 = strongly agree), with higher scores indicating stronger beliefs. Each attitude item was scored on a 7-point Likert scale (1 = strongly dislike, 4 = neither like nor dislike, 7 = strongly like), with higher scores indicating more positive attitudes. For increased clarity, we rescaled responses to [−1,1], so that the indifference point (neither agree nor disagree, neither like nor dislike) was 0. Thus, positive values indicate belief/positive attitudes while negative values indicate disbelief/negative attitudes. We would like to point out that all attitude items were designed as hypothetical statements (e.g. 'I would like if people always had the ability to do otherwise.'). While this did make the items somewhat longer and challenging, we chose this approach to ensure that attitude items did not assume either a belief or a disbelief in the item in question. While 'I like that people always have the ability to do otherwise.' can be a good measure of FWAs in principle, it assumes that people believe free will exists, which might be a problem especially for participants who do not believe so. This problem is avoided when using hypothetical statements, which we chose to do here.

The FWI measures belief in free will (FWB, whether participants think that free will exists). It further measures belief in determinism (DEB, whether participants believe that all events in the physical world are fully determined by prior events), and belief in dualism (DUB, whether participants believe that the mind/soul is a non-physical entity that cannot be reduced to the brain). These latter two beliefs are related to FWBs [11,36] and were measured in this study as well. Sub-scale scores were computed by averaging responses to all items within that sub-scale for each participant. Given that we rephrased each individual FWI item to measure attitudes, we computed sub-scale scores for the attitude items using the same procedure, which yielded separate scores for FWA, determinism attitudes (DEA) and dualism attitudes (DUA).

One key advantage of using this procedure is that both beliefs and attitudes are measured using similar methods, making measures highly comparable. In a previous study, FWAs were measured using a somewhat different approach [26]. Participants were asked whether they felt positively/ negatively, and warm/cold about free will, and whether they liked it or not. This approach is widely used in attitude research since it is simple and intuitive [37]. Yet, while asking participants whether they like free will is indeed somewhat intuitive, that is not the case for determinism and dualism. Many people might not know about the concept of dualism at all and thus could not assess whether or not they like it. For this reason, we decided to deviate from this approach. Instead of directly asking about attitudes toward the underlying constructs (free will, determinism, dualism), we followed the FWI and asked about attitudes toward statements associated with these constructs, assuming that asking participants e.g. 'I would like to live in a world where people always have free will' would be more intuitive and preferable.

### 2.2.2. Post session

**Free will belief manipulation**. Participants who successfully completed the pre session were invited to the post session 14 days after pre session completion. They had 5 days to submit their responses to the post session. Here, they first were submitted to a FWB manipulation, with each participant being randomly assigned to one of two groups: anti-free will (AntiFW) and Control.

The AntiFW group then read an excerpt from Francis Crick's book *The Astonishing Hypothesis* (see electronic supplementary material, Methods 2). This text argues against the existence of free will and has been used successfully to manipulate FWBs in the past [21]. Participants were given at least 5 min to carefully read the text. The instructions read: 'Please take the time to read this text carefully. You will be asked to summarize it later in the experiment. After 5 min, you can proceed to the next screen.'

In the Control group, participants also read an excerpt from the same book (see electronic supplementary material, Methods 2), but this excerpt now introduced readers to the concept of consciousness and had no anti-free will message. Participants were again given at least 5 min to read the text, and received the same instructions on screen.

**Engagement and attention checks**. In order to test whether participants indeed engaged with the texts, they were instructed to write down what they thought were the three main messages of the text immediately after reading it. Then, participants were asked to select one of their three main messages, and describe how it has proven true in their own lives (for a similar procedure, see [10]). Two expert raters (experimenters) independently rated each response as either high or low effort. If both raters consistently rated responses as low effort, that participant was excluded from the sample and replaced. This procedure ensured that all participants included in the sample indeed engaged with the texts and understood their core messages. Eight participants (0.76%) were excluded and replaced this way.

**Free will attitudes and beliefs**. FWAs and FWBs were assessed just like in the pre session, with items presented in the same order.

## 2.3. Planned analyses

All confirmatory and exploratory analyses described here were performed using RStudio (v. 1.3.1093, R v. 4.0.2), and were pre-registered in the stage 1 registered report (https://osf.io/fya89).

### 2.3.1. Hypothesis 1: FWAs and FWBs are related constructs

To test whether FWAs and FWBs are related constructs, we used data from the pre session. First, we computed a partial Pearson correlation between FWB and FWA (using *ppcor::pcor*), while controlling

for the relations with DEB, DUB, DEA and DUA. This allowed us to measure the unique shared variance between FWB and FWA. The same approach was used to compute a partial correlation between DEB and DEA, and between DUB and DUA. We then tested whether the partial correlation coefficient was significantly above zero, which would indicate that FWB and FWA share unique variance and are positively related. We expected to see a moderate positive correlation between FWAs and FWBs.

### 2.3.2. Hypothesis 2: manipulation check

We then performed a manipulation check, in order to assess whether the FWB manipulation was effective. For this purpose, we first computed a difference score by subtracting FWB in the pre session from FWB in the post session ($FWB_{POST} - FWB_{PRE}$) for each participant. This difference score ($FWB_{DIFF}$) indicated how FWBs changed across time. We then computed a two-sample $t$-test to compare $FWB_{DIFF}$ between the two groups (AntiFW, Control), expecting to see lower scores in the AntiFW, as compared with the Control group. For exploratory purposes, we also performed a one-sample $t$-test against 0 for each group, in order to test whether each group showed either an increase or a decrease in FWB.

### 2.3.3. Hypothesis 3: baseline FWBs modulate FWB manipulations

In order to test whether baseline FWBs modulated the effectiveness of the FWB manipulation, we again used difference scores ($FWB_{DIFF} = FWB_{POST} - FWB_{PRE}$). These scores were entered into an ANCOVA (using *rstatix::anova_test*) with the following factors: group (AntiFW, Control), FWB in the pre session ($FWB_{PRE}$) and FWA in the pre session ($FWA_{PRE}$). If the manipulation effect was indeed modulated by baseline FWBs, we would expect to see a significant two-way interaction between group and $FWB_{PRE}$.

### 2.3.4. Hypothesis 4: baseline FWAs modulate FWB manipulations

In order to test whether baseline free will attitudes modulate the effectiveness of the FWBs manipulation, results from the same ANCOVA as in Hypothesis 3 were evaluated. Specifically, we expected to see a significant 2-way interaction between group and $FWA_{PRE}$, if baseline FWAs modulated the strength of FWB manipulations.

### 2.3.5. Hypothesis 5: FWB manipulations also affect FWAs

In order to assess the specificity of the FWB manipulation, we tested whether it also affected FWAs. For this, we performed an analysis that is analogous to Hypothesis 2. We computed a difference score by subtracting FWA in the pre session from FWA in the post session ($FWA_{POST} - FWA_{PRE}$) for each participant. This difference score ($FWA_{DIFF}$) was then entered into a two-sample $t$-test to compare $FWA_{DIFF}$ between the two groups (AntiFW, Control). If the manipulation indeed affected FWAs, we would expect to see a significant difference here as well. As before, we also performed a one-sample $t$-test against 0 for each group, in order to explore whether each group showed either an increase or a decrease in FWA.

### 2.3.6. Exploratory analysis 1: does the FWB manipulation also affect determinism and dualism beliefs?

In many prior experiments using FWB manipulations, effects were only assessed on FWBs directly, i.e. whether they decrease the belief that free will exists. However, it remains unclear whether manipulations indeed have such specific effects, or whether they also affect related beliefs such as dualism or determinism beliefs. To explore this possibility, we performed the same analyses as outlined in Hypothesis 2, only replacing FWB/FWA, with DEB/DEA and DUB/DUA. This allowed us to further assess the specificity of the free will manipulation employed here.

### 2.3.7. Exploratory analysis 2: effect of demographic variables

We also assessed whether any of the effects of the FWB manipulation on either FWBs or FWAs was moderated by demographic variables. To test this, we performed an exploratory ANCOVA with the predictors age, gender, education and strength of religious beliefs and the dependent variable $FWB_{DIFF}$. The same ANCOVA was performed on $FWA_{DIFF}$.

Additionally, we explored whether the same demographic variables affected overall FWBs or FWAs. For this purpose, we tested whether FWB and/or FWA in the pre session could be predicted from either group, age, gender, education or strength of religious beliefs, using an ANCOVA approach.

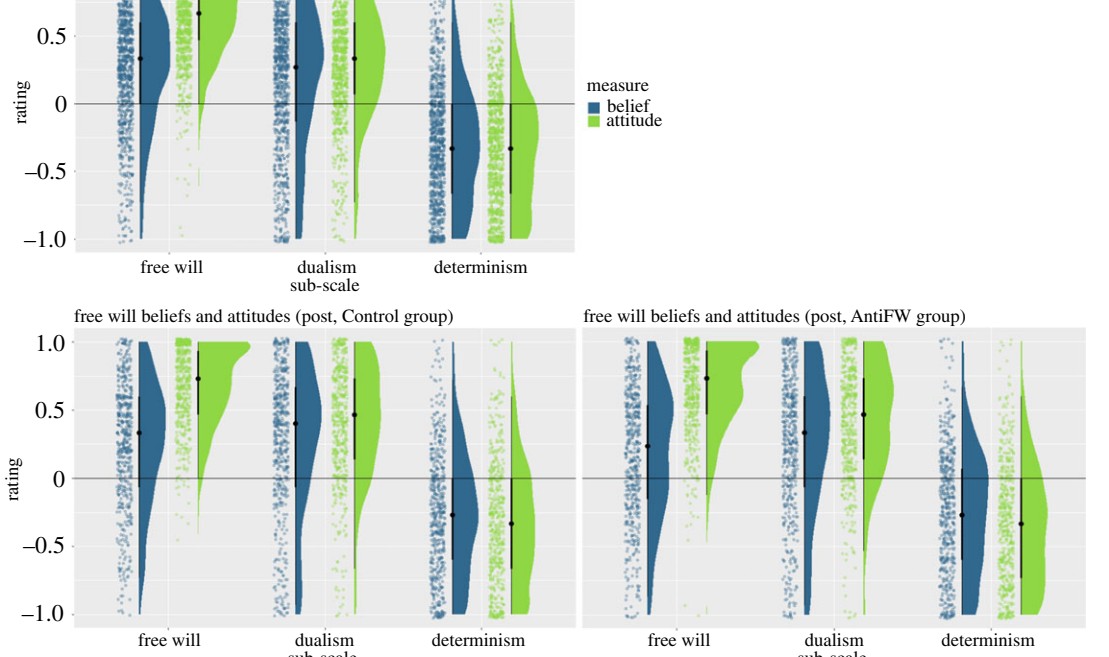

**Figure 1.** Responses to FWB and FWA sub-scales in the pre session (above) and the post session (below, Control group left, Anti-free-will group right). The horizontal line delineates the mid-point of the scale, values above the line indicate belief/positive attitudes, values below the line indicate disbelief/negative attitudes. Each dot represents a single participant, data is jittered for display purposes.

# 3. Results

## 3.1. Mean responses

### 3.1.1. Non-pre-registered analyses

In order to visualize our results, we first plotted responses to all six sub-scales (figure 1, table 1). Overall, in the pre session, participants believed in free will (mean = 0.28, s.d. = 0.45) and in dualism (mean = 0.19, s.d. = 0.52). They did not believe in determinism on average though (mean = −0.28, s.d. = 0.44). All belief sub-scales differed significantly from 0, $ts(1055) > 76.89$, $ps < 0.001$. We found similar results for attitudes. Participants had very positive attitudes toward free will (mean = 0.65, s.d. = 0.31) and dualism (mean = 0.31, s.d. = 0.43), while attitudes toward determinism were somewhat negative (mean = −0.31, s.d. = 0.43). All attitude sub-scales differed significantly from 0, $ts(1055) > 76.19$, $ps < 0.001$. We then tested whether FWBs and attitudes differed using a two-sided paired $t$-test. We found a significant effect, $t_{1055} = 26.77$, $p < 0.001$, Cohen's $d = 0.95$, 95% confidence interval (CI) = [0.86, 1.04], demonstrating that although FWB scores were very high, FWA scores were even higher. At the same time, the variance in FWA scores was somewhat lower numerically than the variance in FWB scores. We found a similar pattern for DUB versus DUA, $t_{1055} = 10.66$, $p < 0.001$, $d = 0.25$, CI = [0.17, 0.34]. For DEB versus DEA we also found a significant effect, $t_{1055} = 3.73$, $p < 0.001$, $d = 0.08$, CI = [−0.01, 0.16], but the effect seems too small numerically to be interpretable.

## 3.2. Hypothesis 1

### 3.2.1. Pre-registered analyses

In order to test whether FWB and free will attitudes (FWA) were related, we performed a partial correlation analysis, controlling for effects of dualism beliefs (DUB) and attitudes (DUA) and determinism beliefs (DEB) and attitudes (DEA). We found FWB and FWA to share unique variance and be positively related, $r = 0.42$, CI = [0.37, 0.47], $p < 0.001$, confirming our hypothesis (figure 2, left panel). Applying the same approach to dualism and determinism, we found DUB and DUA to be

**Table 1.** Descriptive statistics. The mean, median, standard deviation (s.d.), and standard error (s.e.) for each belief (FWB, DUB, DEB) and attitude (FWA, DUA, DEA) sub-scale are shown, separately for the pre session, and both groups in the post session.

| | pre session | | | | post session—Control group | | | | post session—AntiFW group | | | |
|---|---|---|---|---|---|---|---|---|---|---|---|---|
| | mean | median | s.d. | s.e. | mean | median | s.d. | s.e. | mean | median | s.d. | s.e. |
| FWB | 0.28 | 0.33 | 0.45 | 0.01 | 0.25 | 0.33 | 0.46 | 0.02 | 0.18 | 0.2 | 0.49 | 0.02 |
| FWA | 0.65 | 0.67 | 0.31 | 0.01 | 0.68 | 0.73 | 0.3 | 0.01 | 0.65 | 0.73 | 0.33 | 0.01 |
| DUB | 0.19 | 0.27 | 0.52 | 0.02 | 0.27 | 0.4 | 0.53 | 0.02 | 0.24 | 0.33 | 0.53 | 0.02 |
| DUA | 0.31 | 0.33 | 0.43 | 0.01 | 0.4 | 0.47 | 0.43 | 0.02 | 0.42 | 0.47 | 0.42 | 0.02 |
| DEB | −0.39 | −0.3 | 0.44 | 0.01 | −0.26 | −0.27 | 0.45 | 0.02 | −0.25 | −0.27 | 0.45 | 0.02 |
| DEA | −0.32 | −0.33 | 0.43 | 0.01 | −0.33 | −0.33 | 0.46 | 0.02 | −0.34 | −0.33 | 0.44 | 0.02 |

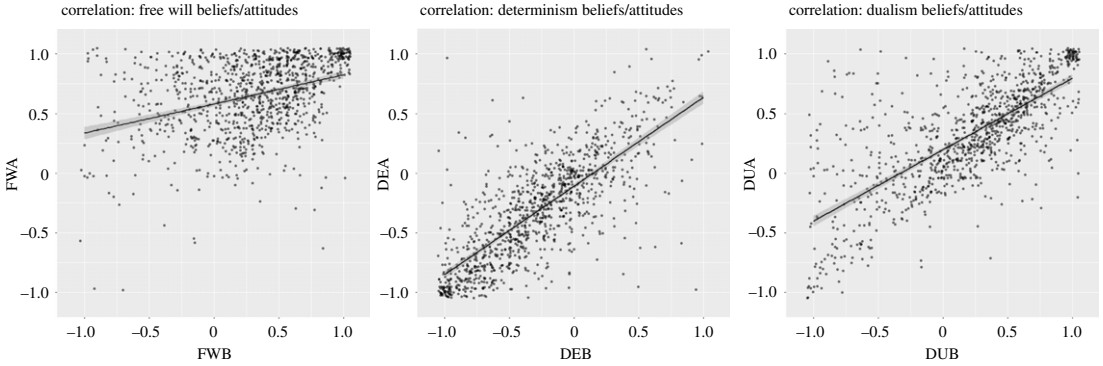

**Figure 2.** Correlations between free will beliefs and attitudes (left), determinism beliefs and attitudes (middle), and dualism beliefs and attitudes (right). Dots represent individual participants, lines are fitted linear functions with 95% confidence intervals (grey areas). Only data from the pre session was used to generate this figure.

**Table 2.** Comparison of belief–attitude correlations. We computed the belief–attitude correlation for each sub-scale, FW = $r$(FWB,FWA), DU = $r$(DUB, DUA), DE = $r$(DEB, DEA), and then computed difference scores between sub-scales (difference). These differences were assessed statistically using *cocor::cocor*, and both $z$-values and $p$-values are reported.

| comparison | difference | z | p |
|---|---|---|---|
| FW–DU | 0.35 | 12.34 | <0.001 |
| FW–DE | 0.4 | 13.82 | <0.001 |
| DU–DE | 0.04 | 2.2 | 0.03 |

positively related, $r = 0.72$, CI = [0.69, 0.75], $p < 0.001$ (figure 2, right panel), and the same was true for DEB and DEA, $r = 0.74$, CI = [0.71, 0.76], $p < 0.001$ (figure 2, middle panel).

### 3.2.2. Non-pre-registered analyses

Notably, we found partial correlations to be lower for free will ($r = 0.42$, shared variance = 17.64%) than for either determinism ($r = 0.74$, shared variance = 54.76%) or dualism ($r = 0.72$, shared variance = 51.84%). In order to corroborate this result, we assessed these differences statistically (using *cocor::cocor*) to compare the correlation coefficients reported above. We found r(FWB, FWA) to be lower than both r(DUB, DUA) and r(DEB, DEA), $ps < 0.001$ (table 2). We also found a significant difference between r(DUB, DUA) = 0.72 and r(DEB, DEA) = 0.74, $p = 0.03$. This demonstrates that with our large sample size even numerically small differences yielded significant results. Still, the difference between r(FWB, FWA) = 0.42 and r(DUB, DUA) = 0.72/r(DEB, DEA) = 0.74 is substantial numerically, demonstrating that FWBs and attitudes were positively related, but to a lesser degree than corresponding beliefs and attitudes toward determinism and dualism. Note that this difference might be partly explained by a ceiling effect in the FWAs, as the variance in the FWB measure (s.d. = 0.45) was substantially larger than in the FWA measure (s.d. = 0.30). To assess this difference statistically, we performed Levene's test and found variances to differ significantly, $F_{1,2110} = 120.41$, $p < 0.001$. Since the lower variance might limit correlations, we corrected for differences in variance in a *post hoc* analysis (using *psych::correct.cor*). We still observed substantially lower correlations between FWB and FWA ($r = 0.60$) than between DUB/DUA ($r = 0.91$) and DEB/DEA ($r = 0.97$).

A number of participants responded only to the pre but not to the post session ($n = 386$), and we used data from these participants as an independent replication sample for Hypothesis 1. Overall, we found results to be similar, FWB and FWA were positively related, $r = 0.35$, CI = [0.26, 0.43], $p < 0.001$, and the same was true for DUB/DUA, $r = 0.77$, CI = [0.72, 0.81], $p < 0.001$, and DEB/DEA, $r = 0.76$, CI = [0.71, 0.80], $p < 0.001$. As before, the correlations for FW were lower than for DU and DE, $ps < 0.001$. Notably, the difference between DU and DE was not significant in this sample, $p = 0.27$.

## 3.3. Hypothesis 2

### 3.3.1. Pre-registered analyses

In order to test if our manipulation was successful, we computed difference scores between the pre and post sessions (FWB$_{DIFF}$ = FWB$_{POST}$ − FWB$_{PRE}$, figure 3) and tested for group differences. We found a

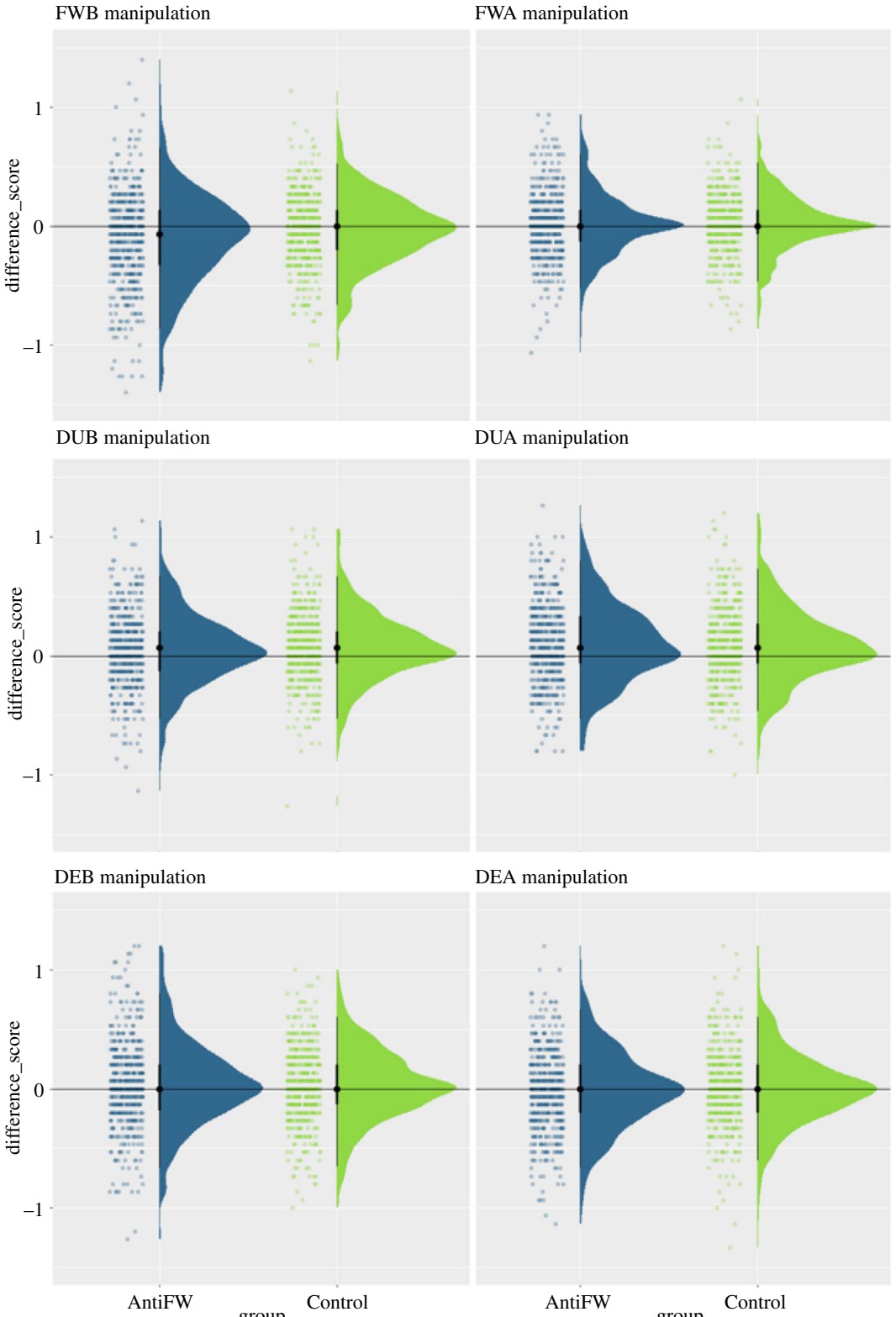

**Figure 3.** Manipulation effects. Free will beliefs (top left), free will attitudes (top right), dualism beliefs (middle left), dualism attitudes (middle right), determinism beliefs (lower left), determinism attitudes (lower right). Positive difference scores indicate an increase across both sessions, negative difference scores indicate a decrease across both sessions.

significant difference, $t_{984.31} = 3.70$, $p < 0.001$, $d = 0.23$, CI = [0.11, 0.35], with scores being lower in the AntiFW group (mean = −0.11), as compared with the Control group (mean = −0.03). We further found the difference scores to be significantly below 0 for the AntiFW group, $t_{527} = -6.31$, $p < 0.001$, $d = -0.27$,

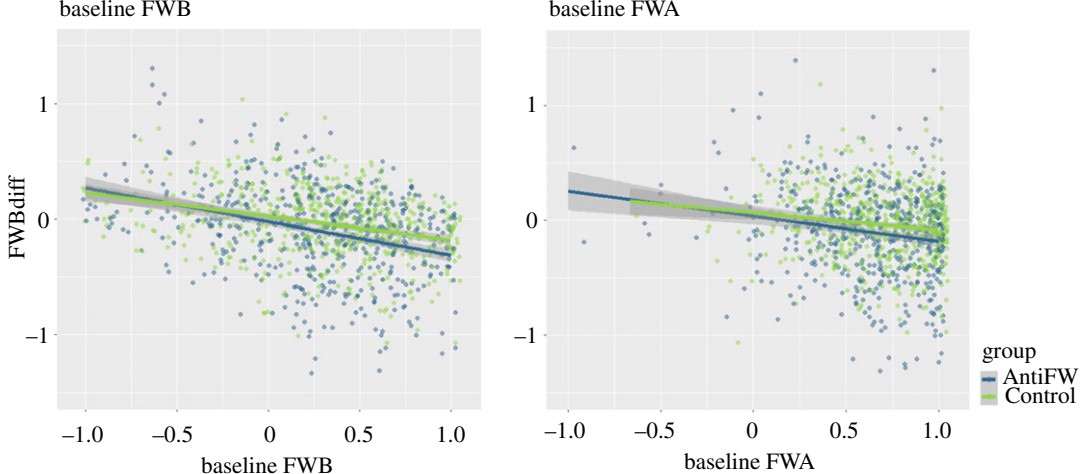

**Figure 4.** Effect of baseline free will beliefs (FWB, left) and baseline free will attitudes (FWA, right) on the free will belief manipulation strength (FWBdiff). AntiFW group is shown in blue, Control group in green, regression lines are plotted with 95% confidence intervals (grey areas).

CI = [−0.45, −0.10]. A similar result was found for the Control group, $t_{527} = -2.18$, $p < 0.001$, $d = -0.09$, CI = [−0.27, 0.08], although the effect was too small numerically to be interpreted meaningfully. Thus, the manipulation worked as expected, reducing FWBs in the AntiFW group, relative to the baseline measure.

### 3.3.2. Non-pre-registered analyses

In order to better compare our results with previous research, we also performed a more traditional comparison of post-manipulation FWB scores across groups, a test that is commonly used in FWB research [18,19,21,28,38]. We found a significant difference, $t_{1049.8} = 2.44$, $p = 0.01$, $d = 0.15$, CI = [0.03, 0.27]. Interestingly, the effect size here was smaller than for the pre-registered test reported above, which suggests that including baseline FWB measures might increase the sensitivity and power of our statistical analysis.

## 3.4. Hypothesis 3

### 3.4.1. Pre-registered analyses

After confirming that our FWB manipulation effectively reduced FWB scores, we tested whether the strength of this effect was modulated by baseline FWBs. For this purpose, we computed an ANCOVA, predicting FWB difference scores ($FWB_{DIFF}$) from baseline FWBs ($FWB_{PRE}$), baseline free will attitudes ($FWA_{PRE}$) and group. We expected to see a group × $FWB_{PRE}$ interaction, which would indicate that group differences were modulated by baseline FWBs. We found a significant main effect of group, $F_{1,1048} = 14.59$, $p < 0.001$, $\eta^2 = 0.01$, indicating a stronger reduction in FWBs in the AntiFW group, as compared with the Control group. We also found a significant main effect of $FWB_{PRE}$, $F_{1,1048} = 100.86$, $p < 0.001$, $\eta^2 = 0.09$, indicating that overall, stronger baseline FWBs led to a stronger reduction in FWBs regardless of group. This effect should be interpreted with care, however, as it might reflect a simple regression-to-the-mean effect. Crucially, we found no signification group × $FWB_{PRE}$ interaction, $F_{1,1048} = 2.66$, $p = 0.104$, $\eta^2 = 0.003$ (figure 4). We can thus conclude that baseline FWBs did not affect the strength of our experimental manipulation.

### 3.4.2. Non-pre-registered analyses

In order to explore the relationship between baseline beliefs and manipulation strength further, we performed a number of additional exploratory analyses. First, while screening the attention check data for each participant, where participants were asked to summarize the main points of the Crick text, we found that some participants ($n = 56$) clearly did not believe the AntiFW text. These participants explicitly stated that the author of the text was wrong and that they believed in free will nonetheless.

As this was not included as an exclusion criterion in the pre-registration, we decided to explore FWB manipulation strength for these participants in an exploratory analysis. We first compared the manipulation strength between this subset of participants and the remaining participants in the AntiFW group, using a two-sample $t$-test. While FWBs decreased for most participants in the AntiFW group ($FWB_{DIFF} = -0.39$), they increased for the subset of participants that did not believe the text ($FWB_{DIFF} = 0.14$), and this difference was significant, $t_{70.89} = -3.26$, $p = 0.001$. Next, we tested whether the effect of baseline beliefs was significant after removing this subset of participants from the sample. Although the $FWB_{PRE} \times$ group interaction effect was slightly stronger, it still did not reach significance, $F_{1,992} = 3.48$, $p = 0.06$, $\eta^2 = 0.003$.

## 3.5. Hypothesis 4

### 3.5.1. Pre-registered analyses

The same ANCOVA which we used for testing Hypothesis 3 was then used to test whether baseline attitudes ($FWA_{PRE}$) also affected the strength of our FWB manipulation. This was again tested by assessing the interaction between group and $FWA_{PRE}$, and this effect was not significant, $F_{1,1048} = 0.28$, $p = 0.56$, $\eta^2 < 0.001$ (figure 4). The same was true for the main effect of $FWA_{PRE}$, $F_{1,1048} = 1.87$, $p = 0.17$, $\eta^2 = 0.002$. Thus, baseline free will attitudes had no detectable effect on manipulation strength.

### 3.5.2. Non-pre-registered analyses

We performed the same exploratory analyses as we did for Hypothesis 3. Removing participants who did not believe the AntiFW manipulation neither led to a significant main effect of baseline attitudes, $F_{1,992} = 0.92$, $p = 0.34$, $\eta^2 = 0.001$, nor to a $FWA_{PRE} \times$ group interaction, $F_{1,992} = 0.02$, $p = 0.88$, $\eta^2 < 0.001$.

## 3.6. Hypothesis 5

### 3.6.1. Pre-registered analyses

After confirming that the FWB manipulation effectively reduced FWB in our sample, we also tested whether a similar effect could be observed on FWA. For this purpose, we compared FWA difference scores between both groups. Counter to our expectations, we found no significant group differences here, $t_{1044.23} = 0.92$, $p = 0.36$, $d = 0.06$, CI $= [-0.06, 0.18]$ (figure 3). In the absence of a significant manipulation effect, we did not perform the planned *post hoc* tests for Hypothesis 5.

## 3.7. Control analyses

### 3.7.1. Non-pre-registered analyses

This study was the first one to use the newly developed attitude version of the FWI, and we, therefore, assessed the psychometric quality of that measure and compared it with the psychometric indicators for the belief version of the FWI. We first assessed the internal consistency of each sub-scale (using *psych:: alpha*, electronic supplementary material, table S1). Although the internal consistencies were slightly lower for attitudes, $\alpha = [0.77–0.79]$ than for beliefs, $\alpha = [0.80–0.85]$, they were still within an acceptable range. Using McDonald's omega led to comparable, slightly higher estimates of internal consistency, $\omega = [0.82–0.88]$ (electronic supplementary material, table S1). Next, we assessed the test-retest reliability of each scale by computing Pearson correlations between the pre and post time points. This analysis was restricted to data from the Control group only, since the manipulation did not strongly affect beliefs or attitudes in this group. Results (electronic supplementary material, table S2) indicated that belief measures showed an acceptable reliability, $r = [0.76–0.85]$. For attitudes, free will showed a slightly lower reliability, $r = 0.65$, CI $= [0.60, 0.70]$, while determinism and dualism showed a higher reliability, $r = 0.75$, CI $= [0.72, 0.79]$, and $r = 0.75$, CI $= [0.71, 0.79]$, respectively. This difference between FWBs and FWAs was probably due to the overserved ceiling effect in FWAs, which reduced variances and, therefore, reduced correlations as well. Lastly, we repeated the confirmatory factor analysis reported in Nadelhoffer *et al*. [17], both for the belief and the attitude versions of the questionnaire (using *lavaan::cfa*). We found acceptable model fit for both the belief version (CFI $= 0.95$, RMSEA $= 0.06$) and the attitude version (CFA $= 0.94$, RMSEA $= 0.05$, for full results see electronic supplementary material, Analysis S1).

## 3.8. Exploratory analysis 1: does the FWB manipulation also affect determinism and dualism beliefs?

### 3.8.1. Pre-registered analyses

One of the main arguments used in the Crick text is that dualism is false, and we, therefore, expected to see manipulation effects on dualism beliefs and attitudes. To test this, we applied the analysis from Hypothesis 2 to both dualism beliefs and attitudes. We neither found a manipulation effect on dualism beliefs, $t_{1049.99} = 1.17$, $p = 0.24$, $d = 0.07$, CI = [−0.05, 0.19], nor on DUA, $t_{1051.19} = −0.84$, $p = 0.40$, $d = −0.05$, CI = [−0.17, 0.07] (figure 2). The same was true for determinism beliefs, $t_{1016.10} = −0.69$, $p = 0.49$, $d = −0.04$, CI = [−0.16, 0.08] and DEA, $t_{1049.27} = 1.03$, $p = 0.30$, $d = 0.06$, CI = [−0.06, 0.18] (figure 2). Thus, counter to our initial expectations, the Crick text had a remarkably specific effect solely on FWBs, while having no effect on related beliefs or attitudes.

### 3.8.2. Non-pre-registered analyses

In figure 2, it can be seen that both dualism beliefs and attitudes increased with time, regardless of group. In order to assess this effect statistically, we computed $t$-tests against 0 for the DUB and DUA, separately for both groups. We found beliefs and attitudes to increase with time in both groups, all $ps < 0.001$, showing a global, unspecific increase in dualism beliefs and attitudes.

## 3.9. Exploratory analysis 2: effect of demographic variables

### 3.9.1. Pre-registered analyses

Lastly, we explored the relation between FWBs and attitudes, and demographic variables in our sample (age, gender, education, strength of religious beliefs). We chose to omit religion type in this analysis, since our sample was dominated by Christian versus non-religious participants, and was thus closely related to the strength of religious beliefs. We found a significant effect of gender on baseline FWBs, $F_{2,1014} = 3.77$, $p = 0.02$, $\eta^2 = 0.007$, which was driven by higher FWBs for male than for female participants, $t_{1041.63} = 1.96$, $p = 0.05$, $d = 0.12$, CI = [0.00, 0.24]. We further found an effect of religious belief strength, $F_{1,1014} = 46.47$, $p < 0.001$, $\eta^2 = 0.04$, showing that stronger religious beliefs were associated with a stronger belief in free will. The interaction between gender and strength of religious beliefs was also significant, $F_{1,1014} = 3.96$, $p = 0.03$, $\eta^2 = 0.007$, showing that gender differences were more pronounced for non-religious participants than they were for religious participants. Lastly, we found no effect of either age, $F_{1,1014} = 1.31$, $p = 0.25$, $\eta^2 = 0.001$, or education, $F_{1,1014} = 0.54$, $p = 0.46$, $\eta^2 = 0.001$.

We then tested whether we could find similar effects on the strength of our free will manipulation (FWB$_{DIFF}$). For this purpose, we computed an ANCOVA with age, gender, education, strength of religious beliefs and group as predictors. Although the effect of religious beliefs did not reach the significance threshold, $F_{1,996} = 3.56$, $p = 0.06$, $\eta^2 = 0.004$, non-religious participants showed a numerically stronger reduction in FWBs than strongly religious participants.

Regarding free will attitudes, we first assessed effects on baseline attitudes (FWA$_{PRE}$). We found a significant effect of age, $F_{1,1014} = 9.19$, $p = 0.002$, $\eta^2 = 0.009$, with older participants showing more positive attitudes toward free will. We also found a significant effect of gender, $F_{2,1014} = 4.15$, $p = 0.01$, $\eta^2 = 0.008$, driven by more positive attitudes for male than female participants, $t_{1030.61} = 2.31$, $p = 0.02$, $d = 0.14$, CI = [0.02, 0.26]. We further found a significant effect of education, $F_{1,1014} = 18.85$, $p < .001$, $\eta^2 = 0.02$, with higher education is being associated with lower free will attitudes. Lastly, the strength of religious beliefs also affected FWA$_{PRE}$, $F_{1,1014} = 9.08$ $p = 0.003$, $\eta^2 = 0.009$, showing that stronger religious beliefs were associated with more negative free will attitudes. Since there was no manipulation effect on free will attitudes, we did not perform the planned analysis linking manipulation strength to demographic variables.

Interestingly, while religious beliefs were associated with increased beliefs in free will, they were also associated with decreased attitudes toward free will. To explore whether this differential effect of religious beliefs on FWBs and attitudes was robust, we performed an additional ANCOVA, predicting ratings from rating type (FWB, FWA) and strength of religious beliefs. A significant interaction would confirm that religious beliefs had differential effects on FWBs and FWAs. Indeed, we found a significant rating type×religious beliefs interaction, $F_{1,2070} = 59.01$, $p < 0.001$, $\eta^2 = 0.03$.

# 4. Discussion

In this registered report, we directly related FWBs and attitudes for the first time. As expected, we found FWBs and FWAs to be positively related, albeit to a lower degree than was the case for either dualism or determinism. We further found a commonly used FWB manipulation (Crick text) to reduce FWB. Counter to our initial predictions, neither baseline beliefs nor baseline attitudes modulated the strength of this manipulation. Furthermore, although we expected the Crick text to also affect dualism beliefs, this was not the case. In fact, the manipulation was remarkably specific to only FWBs, as neither related beliefs, nor attitudes were affected by reading the Crick text.

## 4.1. Free will beliefs versus attitudes

Lay views on free will can be largely separated into beliefs, i.e. agreement to propositional statements about free will, and attitudes, i.e. evaluative judgements about free will [26]. Past research on lay views almost exclusively focused on FWBs, investigating whether FWBs are dynamic and/or consequential [9]. FWBs would be dynamic if they were amenable to change by either internal or external factors (e.g. experimental manipulations), and they would be consequential if they affected downstream behaviour (e.g. social behaviour). While there is considerable doubt whether FWBs are consequential, there is some evidence that they are indeed dynamic and can be changed through experimental manipulations (for a recent meta-analysis, see [21]). By contrast, very little research directly investigated free will attitudes. One previous study established that FWAs are generally positive [26], but used a sub-optimal FWA measure and did not include FWB measures. The current experiment is, therefore, the first one to directly compare FWBs and FWAs.

As expected, we found FWBs and FWAs to be positively related (shared variance = 17.64%), albeit to a lower degree than either DUB/DUA (shared variance = 51.84%) or DEB/DEA (shared variance = 54.76%). One might argue that a ceiling effect and corresponding lower variance in FWAs might explain this difference, yet we found similar results even after correcting for differences in variance. We thus believe that lower FWA variance alone cannot explain this observed difference in belief–attitude relations. One potential explanation for this difference in variance might instead be that participants were more confident of their free will attitudes than they were of their FWBs, leading to more consistent responses. Given the lack of a direct measure of confidence or certainty, this remains speculative at the moment, but future research should directly address this issue by adding within-subjects measures of attitude/belief certainty, instead of merely relying on sample variance.

Overall, the relatively weak correlation between FWBs and FWAs may indicate that both constructs are related, yet separable. This interpretation is corroborated by their relation to religious beliefs. While FWBs were positively related to the strength of religious beliefs, FWAs were negatively related to religious beliefs. Non-religious participants held lower FWBs than strongly religious participants, but also had more positive attitudes toward free will. Although more research will be needed to replicate these findings, demonstrating that a single variable can show opposite correlations with FWBs and FWAs provides evidence for discriminant validity and that both are separable constructs.

## 4.2. Role of baseline beliefs and attitudes

In the past literature, FWB manipulations often showed somewhat inconsistent effects [9], and one of the goals of this registered report was to explain past inconsistencies by identifying modulating factors. Specifically, we assumed that baseline beliefs and attitudes would modulate the strength of experimental manipulations. The more a participant believes in free will, or the more positive their attitude, the more reluctant they should be to change their views after reading a text claiming free will does not exist.

Contrary to our initial expectations, neither of these predictions was supported by our data. Neither baseline FWBs, nor baseline FWAs, measured two weeks before the manipulation was administered, affected its strength. Of course, we cannot rule out that our study was not sufficiently powered to detect a modulation of a small FWB manipulation effect. However, if this effect can only be detected with an even larger sample size ($N > 1056$), it probably is too small to be interpreted in a meaningful way [39]. Thus, it seems more likely that baseline FWBs and FWAs do not affect experimental manipulations. Additionally, we also explored whether demographic variables (age, gender, education, religious beliefs) had a modulating effect on our FWB manipulation, but again found no significant

effects. Note, however, that this study can only speak to the modulation of the Crick manipulation, it might be that other manipulation techniques (e.g. [10]) can be modulated by the variables tested here. In combination with the fact that we only tested a small number of potential modulators, it thus seems too early to conclude that the strength of FWB manipulations cannot be changed in principle.

## 4.3. Specificity of the manipulation

The second goal of this registered report was to assess the specificity of the Crick manipulation, and we expected to see an effect on free will and dualism beliefs as well as free will attitudes. The Crick text primarily attempts to convince participants that dualism is false, and that free will must be non-existent as a consequence. Regardless of whether this argument is sound, reading the Crick text should therefore reduce both free will and dualism beliefs. In fact, this has been demonstrated repeatedly in past research (for a meta-analysis, see [21]). Contrary to our expectations and the past literature, however, we found a remarkably specific effect on FWBs only. Neither dualism beliefs, determinism beliefs, nor any attitude was found to be affected in this experiment.

On the one hand, the fact that only FWBs but not FWAs were affected by the manipulation provides more evidence that these are separable constructs and that evaluative judgements about free will might be harder to change than beliefs (at least using the Crick text). Granted, the Crick manipulation was not originally designed to change FWAs, but it does use negative and somewhat condescending language, which led us to hypothesize it might have an effect on both FWBs and FWAs. This was not the case, and the absence of manipulation effects on FWAs is important for future research investigating free will attitudes. Our results reveal the need to develop novel manipulation strategies specifically targeting FWAs, or to replace experimental methods with correlational approaches if commonly used manipulations remain ineffective.

On the other hand, the fact that only FWBs but not DUBs were affected is at odds with previous research [21], which demonstrated equally strong manipulation effects on free will and dualism beliefs. One potential explanation for this result might be the somewhat different experimental design used here. By contrast to most previous research, which used only a single measurement time point, we chose to also measure baseline beliefs in a separate session two weeks in advance of the manipulation. Although speculative, repeatedly thinking about free will and related constructs might affect responses, for which we found some tentative evidence in our data. Specifically, we found dualism beliefs (and attitudes) to increase with time, regardless of group membership. If thinking repeatedly about dualism increases belief in and attitudes toward dualism, then this might counteract any free will belief reduction that the experimental manipulation aims to achieve, and eliminate manipulation effects. Another potential explanation might be in the sample we acquired. Most of the past research was conducted on student samples, with a bias toward young female participants [9], while we acquired data broadly from a sample with a wide age-range (18–60) and no gender bias (although the sample was still biased toward highly educated participants). Still, acquiring data from student versus non-student samples has had no measurable effects on FWB manipulations in the past [21], and this explanation thus remains tentative at the moment. We remain cautious when interpreting the specificity of the manipulation effect observed here, at least until it can be replicated in future studies.

## 4.4. Relation to religious beliefs

Although this was not the main focus of the current research, we used the large dataset acquired here to explore the relation of free will and religious beliefs. Past research demonstrated that both are intimately linked [40,41], and in an exploratory analysis we investigated whether religious beliefs were related to free will attitudes as well. We found the strength of religious beliefs to be correlated positively with FWBs, with non-religious participants having a weaker belief in free will. Interestingly, we also found the strength of religious beliefs to be correlated negatively with FWAs, with non-religious participants having more positive attitudes toward free will. From a purely methodological viewpoint, this pattern of results demonstrates the discriminant validity of the FWB and FWA measures used here. With the current data, we can only speculate as to why we observed these results. Potentially, Christian participants, which comprised the majority of the religious participants in our sample, believed their free will to be closely linked with responsibility. If this were the case, free will might be perceived as more of a burden. Non-religious participants on the other hand might yearn more for that which they do not believe exists. Additionally, since we only investigated participants from the USA, we cannot rule out that differences between religious and non-religious participants might be specific to this

culture. Results might differ in e.g. European or Asian samples. Overall, these interpretations need to be tested directly in future research.

## 5. Conclusion

In this study, we directly compared FWBs and attitudes for the first time. We found both to be related, yet separable constructs. One main goal was to identify potential moderators for FWB manipulation techniques, potentially explaining why such manipulations often show inconsistent results. However, neither baseline beliefs nor attitudes, nor any demographic variables had a measurable effect on the strength of the FWB manipulation used here. The other main goal was to test the specificity of FWB manipulations. Counter to our expectations and the prior literature, we found these manipulations to show remarkably specific effects solely on FWBs, but not on any other related belief or attitude measured here. Overall, this study provides valuable information for future research on lay views on free will, showing that free will attitudes are separable from FWBs. Thus, free will attitudes might be related to downstream behaviour (e.g. social behaviour), even when FWBs are not.

Ethics. The study was conducted in accordance with the Declaration of Helsinki and has been approved by the Institutional Review Board from the Faculty of Psychology and Educational Science of Ghent University (permit no. 2020/144). All participants provided informed consent at the beginning of the experiment and were informed that participation is voluntary and that all answers will be processed and stored anonymously.

Data accessibility. All materials, data and code are available on the Open Science Framework (https://osf.io/s2vkz/). The pre-registration can also be found on the OSF (https://osf.io/fya89).

The data are provided in electronic supplementary material [42].

Authors' contributions. D.W.: conceptualization, data curation, formal analysis, funding acquisition, investigation, methodology, project administration, resources, software, visualization, writing—original draft, writing—review and editing; E.C.: conceptualization, methodology, software, validation, writing—review and editing; C.G.-G.: conceptualization, methodology, software, validation, writing—review and editing; M.B.: conceptualization, funding acquisition, methodology, writing—review and editing.

All authors gave final approval for publication and agreed to be held accountable for the work performed therein.

Competing interests. The authors declare no competing interests.

Funding. D.W. was supported by the European Union's Horizon 2020 research and innovation programme under the Marie Skłodowska-Curie (MSCA) grant agreement no. 665501 and the Flemish Science Foundation (FWO) grant FWO.KAN.2019.0023.01. E.C. was supported by FWO grants FWO18/PDO/049 and 12U0322N. C.G.-G. was supported by MSCA grant no. 835767 and the Spanish Ministry of Science and Innovation grant IJC2019-040208-I. M.B. was supported by the Einstein Foundation Berlin (Einstein Strategic Professorship).

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
