## [Peer Review File · Royal Society Open Science]

Review History

RSOS-202018.R0 (Original submission)

Review form: Reviewer 1

Do you have any ethical concerns with this paper?

No

Recommendation?

Accept with minor revision

Comments to the Author(s)

Referee Report:

RSOS-202018

1. The research question is scientifically valid.
2. The proposed hypotheses are logical, rational, and plausible.
3. The methodology and analysis pipeline is sound and feasible.
4. The clarity and degree of methodological detail is sufficient to replicate the proposed experimental procedures and analysis pipeline.

5. The authors provide a sufficiently clear and detailed description of the methods to prevent undisclosed flexibility in the experimental procedures or analysis pipeline.
6. The authors have considered sufficient outcome-neutral conditions (e.g. absence of floor or ceiling effects; positive controls; other quality checks) for ensuring that the results obtained are able to test the stated hypotheses.

General Assessment:

A few months ago, I wrote a report for a similar project by the same team of researchers. Here is how I partly summarized that project (which overlaps in this respect with the project under consideration):

“Free will beliefs have recently received a lot of attention in the literature from experimental philosophy, social psychology, and cognitive science. These authors make a novel contribution to this area of research by switching focus from people’s beliefs about free will to their attitudes about free will. As such, this paper represents an important step in a new direction that broadens the scope of research on the role that free will plays in people’s lives. It also opens up a fertile new area for empirical investigation. I nevertheless have several concerns about how attitudes about free will have been operationalized, which in turn problematizes what conclusions we can draw from the findings.”

Several of the concerns I had with that earlier project carry over to the present proposal. But because this research has yet to be done, I feel that the authors really need to address my concerns either by explaining why they are unfounded or by changing their measures accordingly. I believe that the “like” method to measuring attitudes in this context is problematic for the reasons I repeat below. It is something the authors should address before they collect data. To the extent to which they don’t either explain my concerns away or address them, their conclusions about attitudes won’t clearly follow from their findings (since belief will be an obvious confound).

Issues and Suggestions:

First, the authors plan to ask participants to state their like or dislike for statements such as “How people’s lives unfold is completely up to them” (Nadelhoffer et al. 2014). But it’s unclear that liking or disliking this type of statement says anything about how much one values or devalues free will. I might value free will a lot even though I think we lamentably don’t have it. So, how I am supposed to respond given the authors’ methodology? I personally think the statement is false but I wish it were true. So, does that mean that I should dislike it? If so, this merely reflects the fact that I do not *believe* in free will – it says nothing at all about whether I value it. To me, it would make more sense to reword the items from FWI so that the researchers clearly get at attitudes without beliefs as a confound. For instance, “It is important to me [or my worldview] that how people’s lives unfold is completely up to them,” or “I would be upset if scientists proved that how people’s lives unfold isn’t completely up to them.” Something along those lines would unambiguously shed light on attitudes rather than beliefs. The latter option completely addresses the worry I have since someone like me who doesn’t believe in free will but who feels like something valuable is lost in being disillusioned about free will can express the relationship between my beliefs and my attitudes – which is lost given the way the authors propose to measure attitudes. Given that this entire project is about people’s attitudes about free will, I don’t think this is a marginal issue. The items used for measuring these attitudes are crucial. And I worry that some of the items are worded in ways that are either odd and unnatural or that shed light on beliefs and not attitudes (contrary to the intended goal of using these items).

Second, as I pointed out in my previous referee report on one of the authors' earlier projects, there is a related concern when it comes to the relationship between free will beliefs and free will attitudes. If I don't believe that free will exists, then it's hard to know what sense to make of it when I like or dislike a statement that claims we have or lack free will. It would be akin to not believing in souls or ghosts, but saying that I nevertheless like souls or ghosts. If I don't believe that free will exists, what sense does it make for me to either like it or dislike a statement that claims that I have it? Instead, it seems like the issue is whether or not I would prefer to have free will regardless of whether I actually have it. But to get at this, one would need a different approach than the one adopted by the authors. A nested or branching approach would work better. Something like: First, you ask them to agree or disagree with the statements from FWI. If they agree, you ask how much they value it? If they disagree, you ask how much it matters to [or bothers] them that they don't have it. Or something along those lines...

Review form: Reviewer 2

Do you have any ethical concerns with this paper?

No

Recommendation?

Major revision

Comments to the Author(s)

See file (Appendix A).

Review form: Reviewer 3

Do you have any ethical concerns with this paper?

No

Recommendation?

Accept with minor revision

Comments to the Author(s)

In this proposed study, the authors seek a better understanding of lay views of free will by considering both attitudes toward free will and free will beliefs. Past work in this area has exclusively focused on free will beliefs, leaving open the possibility that experimental manipulations have different effects for attitudes than for beliefs or that the degree to which someone values the idea of free will moderates the effect of manipulations on free will beliefs.

The proposed data collection and analysis plan are feasible and sound. The study design uses established measures and manipulations. This research promises to yield valuable insights and help clarify unsettled issues in the literature (cf. Buttrick et al., 2020). While the focus on US adults is consistent with prior work in this area, I suggest adding some discussion and consideration of research showing considerable cross-national variability in the strength and correlates of free will beliefs.

My only criticism of the proposal is that the proposed hypotheses are left rather vague. I suggest revisions to the introduction to clarify links to theory, extant research, and the three research

questions for the five hypotheses listed in the analysis plan (e.g., reframe and expand the discussion on the first full paragraphs of p. 5 and p. 6 to relate to the specific hypotheses to be tested).

Buttrick, N. R., Aczel, B., Aeschbach, L. F., Bakos, B. E., Brühlmann, F., Claypool, H. M., ... & Wood, M. J. (2020). Many Labs 5: Registered Replication of Vohs and Schooler (2008), Experiment 1. *Advances in Methods and Practices in Psychological Science*, 3(3), 429-438.

Decision letter (RSOS-202018.R0)

Dear Dr Wisniewski,

The Editors assigned to your Stage 1 Registered Report ("Relating Free Will Beliefs and Attitudes") have now received comments from reviewers. We would like you to revise your paper in accordance with the referee and editors suggestions which can be found below (not including confidential reports to the Editor).

Please submit a copy of your revised paper within three weeks (i.e. by the 03-Mar-2021).

When submitting your revised manuscript, you must respond to the comments made by the referees and upload a file "Response to Referees" in "Section 2 - File Upload". Please use this to document how you have responded to the comments, and the adjustments you have made. In order to expedite the processing of the revised manuscript, please be as specific as possible in your response.

Kind regards,
Professor Chris Chambers
Royal Society Open Science
openscience@royalsociety.org

on behalf of Professor Chris Chambers (Registered Reports Editor, Royal Society Open Science)
openscience@royalsociety.org

Associate Editor Comments to Author (Professor Chris Chambers):

Associate Editor: 1

Comments to the Author:

Three expert reviewers have now assessed the Stage 1 manuscript. All find varying degrees of merit in the proposal, while also raising concerns with study confounds (especially between valuing and believing), the validity of the experimental intervention, and the precision of the hypotheses. In reading the manuscript, I found myself agreeing that parts of the analysis may be overly complex (and perhaps underspecified) compared to the underlying predictions -- for instance, the ANOVA and ANCOVA analyses will necessarily break down to specific comparisons between conditions. Could it be preferable to the "cut to the case" and target these predictions more directly with specific analyses? I will leave you to thoroughly consider the various comments and recommendations in a revised Stage 1 manuscript.

Comments to Author:

Reviewer: 1

Comments to the Author(s)

Referee Report:

RSOS-202018

1. The research question is scientifically valid.
2. The proposed hypotheses are logical, rational, and plausible.
3. The methodology and analysis pipeline is sound and feasible.
4. The clarity and degree of methodological detail is sufficient to replicate the proposed experimental procedures and analysis pipeline.
5. The authors provide a sufficiently clear and detailed description of the methods to prevent undisclosed flexibility in the experimental procedures or analysis pipeline.
6. The authors have considered sufficient outcome-neutral conditions (e.g. absence of floor or ceiling effects; positive controls; other quality checks) for ensuring that the results obtained are able to test the stated hypotheses.

General Assessment:

A few months ago, I wrote a report for a similar project by the same team of researchers. Here is how I partly summarized that project (which overlaps in this respect with the project under consideration):

"Free will beliefs have recently received a lot of attention in the literature from experimental philosophy, social psychology, and cognitive science. These authors make a novel contribution to this area of research by switching focus from people's beliefs about free will to their attitudes about free will. As such, this paper represents an important step in a new direction that broadens the scope of research on the role that free will plays in people's lives. It also opens up a fertile new area for empirical investigation. I nevertheless have several concerns about how attitudes about free will have been operationalized, which in turn problematizes what conclusions we can draw from the findings."

Several of the concerns I had with that earlier project carry over to the present proposal. But because this research has yet to be done, I feel that the authors really need to address my concerns either by explaining why they are unfounded or by changing their measures accordingly. I believe that the "like" method to measuring attitudes in this context is problematic for the reasons I repeat below. It is something the authors should address before they collect data. To the extent to which they don't either explain my concerns away or address them, their conclusions about attitudes won't clearly follow from their findings (since belief will be an obvious confound).

Issues and Suggestions:

First, the authors plan to ask participants to state their like or dislike for statements such as “How people’s lives unfold is completely up to them” (Nadelhoffer et al. 2014). But it’s unclear that liking or disliking this type of statement says anything about how much one values or disvalues free will. I might value free will a lot even though I think we lamentably don’t have it. So, how I am supposed to respond given the authors’ methodology? I personally think the statement is false but I wish it were true. So, does that mean that I should dislike it? If so, this merely reflects the fact that I do not *believe* in free will – it says nothing at all about whether I value it. To me, it would make more sense to reword the items from FWI so that the researchers clearly get at attitudes without beliefs as a confound. For instance, “It is important to me [or my worldview] that how people’s lives unfold is completely up to them,” or “I would be upset if scientists proved that how people’s lives unfold isn’t completely up to them.” Something along those lines would unambiguously shed light on attitudes rather than beliefs. The latter option completely addresses the worry I have since someone like me who doesn’t believe in free will but who feels like something valuable is lost in being disillusioned about free will can express the relationship between my beliefs and my attitudes – which is lost given the way the authors propose to measure attitudes. Given that this entire project is about people’s attitudes about free will, I don’t think this is a marginal issue. The items used for measuring these attitudes are crucial. And I worry that some of the items are worded in ways that are either odd and unnatural or that shed light on beliefs and not attitudes (contrary to the intended goal of using these items).

Second, as I pointed out in my previous referee report on one of the authors’ earlier projects, there is a related concern when it comes to the relationship between free will beliefs and free will attitudes. If I don’t believe that free will exists, then it’s hard to know what sense to make of it when I like or dislike a statement that claims we have or lack free will. It would be akin to not believing in souls or ghosts, but saying that I nevertheless like souls or ghosts. If I don’t believe that free will exists, what sense does it make for me to either like it or dislike a statement that claims that I have it? Instead, it seems like the issue is whether or not I would prefer to have free will regardless of whether I actually have it. But to get at this, one would need a different approach than the one adopted by the authors. A nested or branching approach would work better. Something like: First, you ask them to agree or disagree with the statements from FWI. If they agree, you ask how much they value it? If they disagree, you ask how much it matters to [or bothers] them that they don’t have it. Or something along those lines...

Reviewer: 2

Comments to the Author(s)

See file

Reviewer: 3

Comments to the Author(s)

In this proposed study, the authors seek a better understanding of lay views of free will by considering both attitudes toward free will and free will beliefs. Past work in this area has exclusively focused on free will beliefs, leaving open the possibility that experimental manipulations have different effects for attitudes than for beliefs or that the degree to which someone values the idea of free will moderates the effect of manipulations on free will beliefs.

The proposed data collection and analysis plan are feasible and sound. The study design uses established measures and manipulations. This research promises to yield valuable insights and help clarify unsettled issues in the literature (cf. Buttrick et al., 2020). While the focus on US adults is consistent with prior work in this area, I suggest adding some discussion and

consideration of research showing considerable cross-national variability in the strength and correlates of free will beliefs.

My only criticism of the proposal is that the proposed hypotheses are left rather vague. I suggest revisions to the introduction to clarify links to theory, extant research, and the three research questions for the five hypotheses listed in the analysis plan (e.g., reframe and expand the discussion on the first full paragraphs of p. 5 and p. 6 to relate to the specific hypotheses to be tested).

Buttrick, N. R., Aczel, B., Aeschbach, L. F., Bakos, B. E., Brühlmann, F., Claypool, H. M., ... & Wood, M. J. (2020). Many Labs 5: Registered Replication of Vohs and Schooler (2008), Experiment 1. *Advances in Methods and Practices in Psychological Science*, 3(3), 429-438.

Author's Response to Decision Letter for (RSOS-202018.R0)

See Appendix B.

RSOS-202018.R1 (Revision)

Review form: Reviewer 1

Do you have any ethical concerns with this paper?

No

Recommendation?

Accept in principle

Comments to the Author(s)

I reviewed this project at an earlier stage of the process. I suggested then that the project was both interesting and potentially ground-breaking. I raised some methodological concerns about the stimuli and measures, and the authors have subsequently adequately addressed my earlier concerns. I think the authors should be invited to collect the proposed data, write up the findings, and publish their paper in RSOS.

- The scientific validity of the research question(s): The research questions are interesting, timely, and valid.
- The logic, rationale, and plausibility of the proposed hypotheses: The hypotheses are straightforward and well-motivated by previous research on free will beliefs.
- The soundness and feasibility of the methodology and analysis pipeline (including statistical power analysis where applicable): The methodology used by the authors is sound and appropriate given their interests. The analysis pipeline is also appropriate.
- Whether the clarity and degree of methodological detail would be sufficient to replicate exactly the proposed experimental procedures and analysis pipeline: Given that the authors are completely transparent, there are no issues where replication is concerned.
- Whether the authors provide a sufficiently clear and detailed description of the methods to prevent undisclosed flexibility in the experimental procedures or analysis pipeline: Everything is described in sufficient detail.

- Whether the authors have considered sufficient outcome-neutral conditions (e.g. positive controls) for ensuring that the results obtained are able to test the stated hypotheses: The proposed experimental design will enable the authors to test their hypotheses.

Review form: Reviewer 2

Do you have any ethical concerns with this paper?

No

Recommendation?

Accept in principle

Comments to the Author(s)

The authors have answered all my comments on the first round of review and I have nothing more to say.

Decision letter (RSOS-202018.R1)

Dear Dr Wisniewski

On behalf of the Editor, I am pleased to inform you that your Manuscript RSOS-202018.R1 entitled "Relating Free Will Beliefs and Attitudes" has been accepted in principle for publication in Royal Society Open Science. The reviewers' comments are included at the end of this email.

You may now progress to Stage 2 and complete the study as approved. Before commencing data collection we ask that you:

- 1) Update the journal office as to the anticipated completion date of your study.
- 2) Register your approved protocol on the Open Science Framework (<https://osf.io/rr>) or other recognised repository, either publicly or privately under embargo until submission of the Stage 2 manuscript. Please note that a time-stamped, independent registration of the protocol is mandatory under journal policy, and manuscripts that do not conform to this requirement cannot be considered at Stage 2. The protocol should be registered unchanged from its current approved state, with the time-stamp preceding implementation of the approved study design.

Following completion of your study, we invite you to resubmit your paper for peer review as a Stage 2 Registered Report. Please note that your manuscript can still be rejected for publication at Stage 2 if the Editors consider any of the following conditions to be met:

- The results were unable to test the authors' proposed hypotheses by failing to meet the approved outcome-neutral criteria.
- The authors altered the Introduction, rationale, or hypotheses, as approved in the Stage 1 submission.

- The authors failed to adhere closely to the registered experimental procedures. Please note that any deviations from the approved experimental procedures must be communicated to the editor immediately for approval, and prior to the completion of data collection. Failure to do so can result in revocation of in-principle acceptance and rejection at Stage 2 (see complete guidelines for further information).
- Any post-hoc (unregistered) analyses were either unjustified, insufficiently caveated, or overly dominant in shaping the authors' conclusions.
- The authors' conclusions were not justified given the data obtained.

We encourage you to read the complete guidelines for authors concerning Stage 2 submissions at <https://royalsocietypublishing.org/rsos/registered-reports#ReviewerGuideRegRep>. Please especially note the requirements for data sharing, reporting the URL of the independently registered protocol, and that withdrawing your manuscript will result in publication of a Withdrawn Registration.

Once again, thank you for submitting your manuscript to Royal Society Open Science and we look forward to receiving your Stage 2 submission. If you have any questions at all, please do not hesitate to get in touch. We look forward to hearing from you shortly with the anticipated submission date for your stage two manuscript.

on behalf of Professor Chris Chambers (Registered Reports Editor, Royal Society Open Science)
openscience@royalsociety.org

Reviewers' comments to Author:

Reviewer: 1

Comments to the Author(s)

I reviewed this project at an earlier stage of the process. I suggested then that the project was both interesting and potentially ground-breaking. I raised some methodological concerns about the stimuli and measures, and the authors have subsequently adequately addressed my earlier concerns. I think the authors should be invited to collect the proposed data, write up the findings, and publish their paper in RSOS.

- The scientific validity of the research question(s): The research questions are interesting, timely, and valid.
- The logic, rationale, and plausibility of the proposed hypotheses: The hypotheses are straightforward and well-motivated by previous research on free will beliefs.
- The soundness and feasibility of the methodology and analysis pipeline (including statistical power analysis where applicable): The methodology used by the authors is sound and appropriate given their interests. The analysis pipeline is also appropriate.
- Whether the clarity and degree of methodological detail would be sufficient to replicate exactly the proposed experimental procedures and analysis pipeline: Given that the authors are completely transparent, there are no issues where replication is concerned.
- Whether the authors provide a sufficiently clear and detailed description of the methods to prevent undisclosed flexibility in the experimental procedures or analysis pipeline: Everything is described in sufficient detail.

- Whether the authors have considered sufficient outcome-neutral conditions (e.g. positive controls) for ensuring that the results obtained are able to test the stated hypotheses: The proposed experimental design will enable the authors to test their hypotheses.

Reviewer: 2

Comments to the Author(s)

The authors have answered all my comments on the first round of review and I have nothing more to say.

Author's Response to Decision Letter for (RSOS-202018.R1)

Appendix C.

RSOS-202018.R2

Review form: Reviewer 1

Is the manuscript scientifically sound in its present form?

Yes

Are the interpretations and conclusions justified by the results?

Yes

Is the language acceptable?

Yes

Do you have any ethical concerns with this paper?

No

Have you any concerns about statistical analyses in this paper?

No

Recommendation?

Accept as is

Comments to the Author(s)

RSOS-202018.R2

Relating Free Will Beliefs and Attitudes

Formal Appraisal

- The data are able to test the authors' proposed hypotheses by passing the approved outcome-neutral criteria (such as absence of floor and ceiling effects or success of positive controls or other quality checks).
- The introduction, rationale and stated hypotheses are the same as the approved Stage1 submission.
- The authors adhered precisely to the registered experimental procedures.

- Any unregistered exploratory statistical analyses are justified, methodologically sound, and informative.
- The authors' conclusions are justified given the data.

Informal Appraisal

Free will beliefs have recently received a lot of attention in the literature from experimental philosophy, social psychology, and cognitive science. These authors make a novel contribution to this area of research by switching focus from people's beliefs about free will to their attitudes about free will. As such, this registered report represents an important step in a new direction that broadens the scope of research on the role that free will plays in people's lives. It also opens up a fertile new area for empirical investigation.

I reviewed this project in two earlier phases of development. The authors have done a satisfactory job addressing any concerns I raised during the previous two rounds of review. Most importantly, the way they eventually designed the items for measuring FWAs avoids the potential confounds I mentioned before and also enables the authors to test their core hypotheses in a straightforward manner. The end result is a registered report that does an excellent job of advancing the literature on how people ordinarily think about free will. In addition to providing new insight into FWBs – which is an important contribution to the extant research in its own right – the authors have also developed a novel method for measuring FWAs, which they convincingly argue should receive more attention.

Finally, the stimuli and measures are all well-designed and well-suited for testing the main hypotheses, the analyses are all appropriate, the findings are novel and important, and the discussion of the findings is clear and concise. For all of these reasons, I believe that this registered report should be published as is.

Review form: Reviewer 2

Is the manuscript scientifically sound in its present form?

Yes

Are the interpretations and conclusions justified by the results?

Yes

Is the language acceptable?

Yes

Do you have any ethical concerns with this paper?

No

Have you any concerns about statistical analyses in this paper?

No

Recommendation?

Accept with minor revision

Comments to the Author(s)

Please see the attached file (Appendix D).

Review form: Reviewer 3

Is the manuscript scientifically sound in its present form?

Yes

Are the interpretations and conclusions justified by the results?

Yes

Is the language acceptable?

Yes

Do you have any ethical concerns with this paper?

No

Have you any concerns about statistical analyses in this paper?

No

Recommendation?

Accept as is

Comments to the Author(s)

This research is important and well-executed. All changes to the Introduction, Rationale and Hypotheses sections were superficial (e.g., changed to past tense) and provide added clarity while not deviating from the Stage 1 submission. New exploratory analyses (pp. 13-14, note 1, pp. 17-18, p. 19, pp. 21-23) are clearly described and sufficiently justified, while providing helpful and informative robustness checks to rule out ceiling effects, etc. The Authors' conclusions are justified and sound, give serious consideration to alternative explanations, and are presented as tentative when appropriate, particularly when findings are contrary to past studies or when additional research is warranted.

Decision letter (RSOS-202018.R2)

Dear Dr Wisniewski:

On behalf of the Editor, I am pleased to inform you that your Stage 2 Registered Report RSOS-202018.R2 entitled "Relating Free Will Beliefs and Attitudes" has been deemed suitable for publication in Royal Society Open Science subject to minor revision in accordance with the referee suggestions. Please find the referees' comments at the end of this email.

The reviewers and Subject Editor have recommended publication, but also suggest some minor revisions to your manuscript. We invite you to respond to the comments and revise your manuscript. Below the referees' and Editors' comments (where applicable) we provide additional requirements. Final acceptance of your manuscript is dependent on these requirements being met. We provide guidance below to help you prepare your revision.

Please submit your revised manuscript and required files (see below) no later than 21 days from today's (ie 21-Dec-2021) date. Note: the ScholarOne system will 'lock' if submission of the revision is attempted after the deadline. If you do not think you will be able to meet this deadline please contact the editorial office immediately.

on behalf of Professor Chris Chambers
(Registered Reports Editor, Royal Society Open Science)
openscience@royalsociety.org

Associate Editor Comments to Author (Professor Chris Chambers):

Associate Editor: 1

Comments to the Author:

The three reviewers who participated at Stage 1 kindly returned to evaluate the Stage 2 submission. As you will see, all are very positive, with Reviewers 1 and 3 recommending acceptance as is. I concur with these sentiments and very much enjoyed reading your Stage 2 manuscript. Reviewer 2, however, offers what I think are some useful constructive suggestions for improving clarity and insight, primarily in the reporting of certain aspects of the statistics and in the Discussion section. A minor revision is therefore warranted. In revising, please avoid making any further changes to the Introduction and Method sections, unless doing so is necessary to correct factual errors, grammatical or typographical errors, or to resolve points of confusion. I will assess the revised version at desk, and final acceptance should be forthcoming without requiring further in-depth review.

Comments to Author:

Reviewer: 1

Comments to the Author(s)

RSOS-202018.R2

Relating Free Will Beliefs and Attitudes

Formal Appraisal

- The data are able to test the authors' proposed hypotheses by passing the approved outcome-neutral criteria (such as absence of floor and ceiling effects or success of positive controls or other quality checks).
- The introduction, rationale and stated hypotheses are the same as the approved Stage1 submission.

- The authors adhered precisely to the registered experimental procedures.
- Any unregistered exploratory statistical analyses are justified, methodologically sound, and informative.
- The authors' conclusions are justified given the data.

Informal Appraisal

Free will beliefs have recently received a lot of attention in the literature from experimental philosophy, social psychology, and cognitive science. These authors make a novel contribution to this area of research by switching focus from people's beliefs about free will to their attitudes about free will. As such, this registered report represents an important step in a new direction that broadens the scope of research on the role that free will plays in people's lives. It also opens up a fertile new area for empirical investigation.

I reviewed this project in two earlier phases of development. The authors have done a satisfactory job addressing any concerns I raised during the previous two rounds of review. Most importantly, the way they eventually designed the items for measuring FWAs avoids the potential confounds I mentioned before and also enables the authors to test their core hypotheses in a straightforward manner. The end result is a registered report that does an excellent job of advancing the literature on how people ordinarily think about free will. In addition to providing new insight into FWBs – which is an important contribution to the extant research in its own right – the authors have also developed a novel method for measuring FWAs, which they convincingly argue should receive more attention.

Finally, the stimuli and measures are all well-designed and well-suited for testing the main hypotheses, the analyses are all appropriate, the findings are novel and important, and the discussion of the findings is clear and concise. For all of these reasons, I believe that this registered report should be published as is.

Reviewer: 2

Comments to the Author(s)
Please see the Attached file

Reviewer: 3

Comments to the Author(s)

This research is important and well-executed. All changes to the Introduction, Rationale and Hypotheses sections were superficial (e.g., changed to past tense) and provide added clarity while not deviating from the Stage 1 submission. New exploratory analyses (pp. 13-14, note 1, pp. 17-18, p. 19, pp. 21-23) are clearly described and sufficiently justified, while providing helpful and informative robustness checks to rule out ceiling effects, etc. The Authors' conclusions are justified and sound, give serious consideration to alternative explanations, and are presented as tentative when appropriate, particularly when findings are contrary to past studies or when additional research is warranted.

===PREPARING YOUR MANUSCRIPT===

one version should clearly identify all the changes that have been made (for instance, in coloured highlight, in bold text, or tracked changes);
 a 'clean' version of the new manuscript that incorporates the changes made, but does not highlight them. This version will be used for typesetting.

===PREPARING YOUR REVISION IN SCHOLARONE===

-- If you are requesting an article processing charge waiver, you must select the relevant waiver option (if requesting a discretionary waiver, the form should have been uploaded, see 'File upload' above).

-- If you have uploaded any electronic supplementary (ESM) files, please ensure you follow the guidance at <https://royalsociety.org/journals/authors/author-guidelines/#supplementary-material> to include a suitable title and informative caption. An example of appropriate titling and captioning may be found at https://figshare.com/articles/Table_S2_from_Is_there_a_trade-off_between_peak_performance_and_performance_breadth_across_temperatures_for_aerobic_sc_ope_in_teleost_fishes_/3843624.

Author's Response to Decision Letter for (RSOS-202018.R2)

See Appendix E.

Decision letter (RSOS-202018.R3)

Dear Dr Wisniewski:

It is a pleasure to accept your manuscript entitled "Relating Free Will Beliefs and Attitudes" in its current form for publication in Royal Society Open Science.

Please ensure that you send to the editorial office an editable version of your accepted manuscript - failure to provide this may delay the processing of your proof.

Thank you for your fine contribution. On behalf of the Editors of Royal Society Open Science, we look forward to your continued contributions to the journal.

on behalf of Professor Chris Chambers (Subject Editor)
openscience@royalsociety.org

Appendix A

Review of the registered report of “Relating Free Will Beliefs and Attitudes”

Thank you for the reading of this registered report that I find very interesting and a true improvement for the field of research in folk’s understanding of free will.

As stated in the introduction, most studies in the field are about belief in free will and not attitudes toward free will. Also, as far as I know, no studies relate beliefs and attitudes and I find it a really important contribution.

I have a concern regarding the overall methodology. Everything is based upon the manipulation of Vohs and Schooler (2008). We have many examples that this study has problems. Manylab5 found out that the manipulation just... does not work (Buttrick et al., 2020¹). Another recent study did not replicate the results (Nadelhoffer et al., 2020²) and researchers concluded that “We suggest that manipulating free will beliefs in a robust way is more difficult than has been implied by prior work”. I also used Vohs and Schooler (2008) manipulation in a related decision-making task and found nothing, (while I had a n = 122). To be clear, at this moment, the most probable evidence is that Vohs and Schooler (2008) manipulation simply does not work.

It is a big concern concerning this registered report because if the manipulation does not work, as you stated in table 1, it implies that: “The free will belief manipulation (a) has no effect (b) has an effect that is too small to detect given the statistical power of this study”. There is simply no alternative explanation to this and everything would be a waste of data if the manipulation does not work (and it most probably will).

A second concern I have is about the questions asked to the participants. Please note that I am not an expert in the difference between belief and attitude but I try to imagine myself as a participant. If I had to answers the same questions with the same scales with a subtle difference of framing in the

¹ <https://doi.org/10.1177/2515245920917931>

² <https://doi.org/10.1016/j.cognition.2020.104342>

response between “agree/disagree” and “like/dislike” for 15 questions (30 in total), I am not sure I will understand the differences.

While I understand the argument of intuitiveness, I think that for such subtle differences as between belief and attitude, the questions asked to the participants must be very clear in their mind: “do I think this idea is true”, and “do I like this idea”. I am not sure that, with your design, participants have a clear understanding of this difference. I don’t have a clear and helpful argument to give but I don’t want data from 1100 participants going to waste because we did not pre-test the questions or ensure that they (and their differences) are well understood by the participants.

My third concern is about data analysis. In the references I provided, the effect sizes are so subtle and so weak that ANOVA and ANCOVA will always overestimate the effects. I suggest you move to more straightforward and direct statistics. What we want is a correlation between belief and attitudes and just it would be a strong contribution to the field. The more analyses we do, the more significant results we can find (i.e., possibility of false positive) and that is exactly what we want to avoid in a registered report. Depending on where you move your design, you probably will find ways to find meaningful results while simplifying your analysis.

Here is a summary of my suggestions:

1. Change the design of the study by excluding manipulation by Vohs and Schooler, since we are now more and more sure that it does not work. Go for a correlational design between belief and attitude OR design a backup plan in case the manipulation does not work to not waste the data collection. Make things simple.
2. Find a way to ensure that the questions asked are well understood. Think about the number of questions: is it worth having 15 questions for only one construct? Would having only one more straightforward question that we are sure is well understood not better? How can you be sure?

3. Simplify your analysis. You are trying to find a correlation and a way to “decorrelate” two constructs. You don’t need many anovas and ancovas, and, as I pointed out, the more analyses you conduct, the more you risk finding false-positive results and we want to avoid that at all cost, especially in registered reports.

On the overall, I think that a study of the relationship between belief and attitude concerning free will is a very important contribution to the field. I am not convinced that this design and analysis can provide valuable information concerning this relationship and proposed some suggestions to enhance the design and analysis.

Best regards,

Adrien Fillon

Appendix B

Editor:

Three expert reviewers have now assessed the Stage 1 manuscript. All find varying degrees of merit in the proposal, while also raising concerns with study confounds (especially between valuing and believing), the validity of the experimental intervention, and the precision of the hypotheses. In reading the manuscript, I found myself agreeing that parts of the analysis may be overly complex (and perhaps underspecified) compared to the underlying predictions -- for instance, the ANOVA and ANCOVA analyses will necessarily break down to specific comparisons between conditions. Could it be preferable to the "cut to the case" and target these predictions more directly with specific analyses? I will leave you to thoroughly consider the various comments and recommendations in a revised Stage 1 manuscript.

We thank the editor for the work they put into evaluating our work. We agree that several aspects of the proposed research still require some work, and we heavily revised most parts of the manuscript. This includes addressing confounds, issues with the manipulation, and complexity of the analyses. Please find our point by point responses below. Given the many changes, we provide both a version of the manuscript with all changes marked in red, as well as a version without marked changes.

Reviewer 1:

A few months ago, I wrote a report for a similar project by the same team of researchers. Here is how I partly summarized that project (which overlaps in this respect with the project under consideration):

"Free will beliefs have recently received a lot of attention in the literature from experimental philosophy, social psychology, and cognitive science. These authors make a novel contribution to this area of research by switching focus from people's beliefs about free will to their attitudes about free will. As such, this paper represents an important step in a new direction that broadens the scope of research on the role that free will plays in people's lives. It also opens up a fertile new area for empirical investigation. I nevertheless have several concerns about how attitudes about free will have been operationalized, which in turn problematizes what conclusions we can draw from the findings."

Several of the concerns I had with that earlier project carry over to the present proposal. But because this research has yet to be done, I feel that the authors really need to address my concerns either by explaining why they are unfounded or by changing their measures accordingly. I believe that the "like" method to measuring attitudes in this context is problematic for the reasons I repeat below. It is something the authors should address before they collect data. To the extent to which they don't either explain my concerns away or address them, their conclusions about attitudes won't clearly follow from their findings (since belief will be an obvious confound).

First, we would like to thank the reviewer for agreeing to evaluate this follow-up proposal to our original study. We highly appreciate the valuable comments that are informed by our prior submission, this type of interaction is quite rare and very helpful.

Issues and Suggestions:

First, the authors plan to ask participants to state their like or dislike for statements such as “How people’s lives unfold is completely up to them” (Nadelhoffer et al. 2014). But it’s unclear that liking or disliking this type of statement says anything about how much one values or devalues free will. I might value free will a lot even though I think we lamentably don’t have it. So, how I am supposed to respond given the authors’ methodology? I personally think the statement is false but I wish it were true. So, does that mean that I should dislike it? If so, this merely reflects the fact that I do not **believe** in free will—it says nothing at all about whether I value it. To me, it would make more sense to reword the items from FWI so that the researchers clearly get at attitudes without beliefs as a confound. For instance, “It is important to me [or my worldview] that how people’s lives unfold is completely up to them,” or “I would be upset if scientists proved that how people’s lives unfold isn’t completely up to them.” Something along those lines would unambiguously shed light on attitudes rather than beliefs. The latter option completely addresses the worry I have since someone like me who doesn’t believe in free will but who feels like something valuable is lost in being disillusioned about free will can express the relationship between my beliefs and my attitudes—which is lost given the way the authors propose to measure attitudes. Given that this entire project is about people’s attitudes about free will, I don’t think this is a marginal issue. The items used for measuring these attitudes are crucial. And I worry that some of the items are worded in ways that are either odd and unnatural or that shed light on beliefs and not attitudes (contrary to the intended goal of using these items).

We fully agree with the reviewer that the operationalization of our free will attitude measures is central to the proposed experiment. And as reviewer #2 pointed out as well, the current operationalization is not optimal. For one, the belief and attitude items are identical, making it difficult for participants to differentiate their beliefs from their attitudes.

We thus followed the reviewers’ suggestion and reformulated the attitude items such that they unambiguously reflect attitudes instead of beliefs. For this purpose, the FWI items have been altered to be hypothetical (e.g. “If it were up to me...”, “I would like if...”), and always express a liking judgment. Responses will be given on a liking scale (7-point Likert scale, 1 = strongly dislike, 4 = neither like nor dislike, 7 = strongly like). These revised measures ensure that beliefs and attitudes can be easily distinguished and better dissociated, allowing participants to express an attitude that differs from their belief (“I do not believe this, but I like the idea nonetheless”).

Additionally, we will change the order in which the items will be presented. Instead of presenting all belief items first, and then presenting all attitude items, we will present the same item first in its original / belief form, and then in its attitude form. We hope that this will emphasize the difference between its two forms even more.

Here are the items we will use. On the left you see the FWI items in the original form, on the right the attitude versions. If the reviewer has any further feedback on the specific wording of any of the items we plan to use, it is obviously highly welcome since this is central to the planned research.

Free will

Beliefs	Attitudes
People always have the ability to do otherwise.	I would like if people always had the ability to do otherwise.
People always have free will.	I would like to live in a world where people always have free will.
How people's lives unfold is completely up to them.	I would like to live in a world where how people's lives unfold would be completely up to them.
People ultimately have complete control over their decisions and their actions.	If it were up to me, I would like people to ultimately have complete control over their decisions and their actions.
People have free will even when their choices are completely limited by external circumstances.	If it were up to me, I would like people to have free will even when their choices are completely limited by external circumstances.

Determinism

Normal	Attitudes
Everything that has ever happened had to happen precisely as it did, given what happened before.	I would like to live in a world where everything that has ever happened had to happen precisely as it did, given what happened before.
Every event that has ever occurred, including human decisions and actions, was completely determined by prior events.	If it were up to me, I would like if every event that has ever occurred, including human decisions and actions, would be completely determined by prior events.
People's choices and actions must happen precisely the way they do because of the laws of nature and the way things were in the distant past.	I would like to live in a world where people's choices and actions must happen precisely the way they do because of the laws of nature and the way things were in the distant past.
A supercomputer that could know everything about the way the universe is now could know everything about the way the universe will be in the future.	If it were up to me, I would like if a supercomputer that could know everything about the way the universe is now could know everything about the way the universe will be in the future.
Given the way things were at the Big Bang, there is only one way for everything to happen in the universe after that.	I would like to live in a world where there was only one way for everything to happen in the universe, given the way things were at the Big Bang.

Dualism

Normal	Attitudes
The fact that we have souls that are distinct from our material bodies is what makes humans unique.	If it were up to me, I would like it what made humans unique were the fact that we have souls that are distinct from our material bodies.
Each person has a non-physical essence that makes that person unique.	I would like to live in a world where each person had a non-physical essence that makes that person unique.
The human mind cannot simply be reduced to the brain.	If it were up to me, I would like if the human mind could not simply be reduced to the brain.
The human mind is more than just a complicated biological machine.	I would like to live in a world where the human mind was more than just a complicated biological machine.
Human action can only be understood in terms of our souls and minds and not just in terms of our brains.	If it were up to me, I would like to live in a world where human action could only be understood in terms of our souls and minds and not just in terms of our brains.

Second, as I pointed out in my previous referee report on one of the authors' earlier projects, there is a related concern when it comes to the relationship between free will beliefs and free will attitudes. If I don't believe that free will exists, then it's hard to know what sense to make of it when I like or dislike a statement that claims we have or lack free will. It would be akin to not believing in souls or ghosts, but saying that I nevertheless like souls or ghosts. If I don't believe that free will exists, what sense does it make for me to either like it or dislike a statement that claims that I have it? Instead, it seems like the issue is whether or not I would prefer to have free will regardless of whether I actually have it. But to get at this, one would need a different approach than the one adopted by the authors. A nested or branching approach would work better. Something like: First, you ask them to agree or disagree with the statements from FWI. If they agree, you ask how much they value it? If they disagree, you ask how much it matters to [or bothers] them that they don't have it. Or something along those lines...

We believe that the rephrasing of the attitude items largely solves this issue. By using hypothetical statements, we unlink attitude items explicitly from any specific belief, making it easier to express diverging beliefs/attitudes. For instance, participants can easily express their disbelief in "People always have the ability to do otherwise." while simultaneously expressing a positive attitude towards "I would like if people always had the ability to do otherwise."

Reviewer 2

Thank you for the reading of this registered report that I find very interesting and a true improvement for the field of research in folk's understanding of free will. As stated in the introduction, most studies in the field are about belief in free will and not attitudes toward free will. Also, as far as I know, no studies relate beliefs and attitudes and I find it a really important contribution.

We thank the reviewer for his overall positive evaluation of our report.

I have a concern regarding the overall methodology. Everything is based upon the manipulation of Vohs and Schooler (2008). We have many examples that this study has problems. Manylab5 found out that the manipulation just... does not work (Buttrick et al., 2020). Another recent study did not replicate the results (Nadelhoffer et al., 2020) and researchers concluded that “We suggest that manipulating free will beliefs in a robust way is more difficult than has been implied by prior work”. I also used Vohs and Schooler (2008) manipulation in a related decision-making task and found nothing, (while I had a $n = 122$). To be clear, at this moment, the most probable evidence is that Vohs and Schooler (2008) manipulation simply does not work.

It is a big concern concerning this registered report because if the manipulation does not work, as you stated in table 1, it implies that: “The free will belief manipulation (a) has no effect (b) has an effect that is too small to detect given the statistical power of this study”. There is simply no alternative explanation to this and everything would be a waste of data if the manipulation does not work (and it most probably will).

We understand the reviewer’s concern here, given some recent failed replications it might seem likely that the manipulation will fail here as well. And if that were the case, we would be able to test hypotheses 1-2, but not Hypotheses 3-5. We do believe however that a free will belief manipulation using the method proposed here is possible. To that end, we recently demonstrated in a large meta-analysis that although the effects are small, free will belief manipulations can change beliefs, even after correcting for publication bias (<https://psyarxiv.com/quwgr/>). These findings give us confidence that a sufficiently highly powered study, such as this one, will be able to detect manipulation effects. Our analysis plan and power analysis were designed with this meta-analysis in mind. The power analysis specifically has been based on results from the meta-analysis, and will allow us to detect the small effects we expect to see. This is why we will need and acquire data from a large sample of $n = 528$ per group, which is considerably larger than in most prior research.

As with any a priori power analysis, this cannot guarantee that the manipulation check will be successful, but it does make it highly likely. In case we see no significant results in the manipulation check, we can still report results from Hypothesis 1 and from the manipulation check, unaltered. For Hypotheses 3-5, the situation looks different, as these rest on the assumption that the manipulation is successful. To hedge against the risk of a failed manipulation check, we devised an alternative analysis plan for this case.

First, if the manipulation failed, we will pool data from both groups ($n = 1056$). This will simplify the design by removing the group factor. Given that the original Hypotheses 3-5 revolved around explaining how manipulation effects are modulated by e.g. FWBs, it makes no sense to try and translate these Hypotheses to a design without experimental groups. Instead, we can focus on assessing the directionality of the relation between FWAs and FWBs. Do beliefs affect attitudes, do attitudes affect beliefs, or both?

To test this, we will leverage the fact that we acquire both FWAs and FWBs on two separate time points. We will use linear models to predict FWB_{post} from FWA_{pre} and FWB_{pre} . If we found a significant effect of FWA_{pre} , this would demonstrate that attitudes acquired at a specific time point predict beliefs at a later time point, while controlling for the predictive effect of beliefs (Hypothesis 6). We can also use this logic

to test for the opposite effect of beliefs on attitudes by estimating a linear model that predicts FWA_{post} from FWA_{pre} and FWB_{pre} (Hypothesis 7).

Thus, even in the unlikely case of a failed manipulation check, we will still be able to show whether FWBs and FWAs are related (Hypothesis 1), whether FWAs predict FWBs (Hypothesis 6), whether FWBs predict FWAs (Hypothesis 7), or both. This set of alternative analyses will therefore still provide valuable information for the literature.

A second concern I have is about the questions asked to the participants. Please note that I am not an expert in the difference between belief and attitude but I try to imagine myself as a participant. If I had to answers the same questions with the same scales with a subtle difference of framing in the response between “agree/disagree” and “like/dislike” for 15 questions (30 in total), I am not sure I will understand the differences.

While I understand the argument of intuitiveness, I think that for such subtle differences as between belief and attitude, the questions asked to the participants must be very clear in their mind: “do I think this idea is true”, and “do I like this idea”. I am not sure that, with your design, participants have a clear understanding of this difference. I don’t have a clear and helpful argument to give but I don’t want data from 1100 participants going to waste because we did not pre-test the questions or ensure that they (and their differences) are well understood by the participants.

We agree that the belief and attitude items are too similar in its current form. Reviewer #1 raised a similar point, and we will take several measures to address this issue.

We will rephrase the FWI items, in order to make them unambiguously about attitudes. They will be altered to hypotheticals (e.g. “If it were up to me...”, “I would like if...”), and always express a liking judgment. Responses will be given on a liking scale (7-point Likert scale, 1 = strongly dislike, 4 = neither like nor dislike, 7 = strongly like). For a complete list of the new items, please see the table above or Appendix 2 in the manuscript. Additionally, we will change the order in which the items will be presented. Instead of presenting all belief items first, and then presenting all attitude items, we will present the same item first in its original / belief form, and then in its attitude form. We hope that this will emphasize the difference between its two item forms even more. These measures ensure that beliefs and attitudes can be easily dissociated.

My third concern is about data analysis. In the references I provided, the effect sizes are so subtle and so weak that ANOVA and ANCOVA will always overestimate the effects. I suggest you move to more straightforward and direct statistics. What we want is a correlation between belief and attitudes and just it would be a strong contribution to the field. The more analyses we do, the more significant results we can find (i.e., possibility of false positive) and that is exactly what we want to avoid in a registered report. Depending on where you move your design, you probably will find ways to find meaningful results while simplifying your analysis.

If we understand the reviewer correctly, he is concerned about both model complexity and number of test we plan to perform in this registered report. He correctly points out that the main goal is to test whether FWBs and FWAs are correlated, and we would like to point out that this test is already included

in the analysis plan (Hypothesis 1), using the simplest possible test (correlation analysis). All additional hypotheses are included to deliver more nuance to the relationship between beliefs and attitudes.

Regarding the number of analyses we plan to perform, we believe that we are already using the minimum number of tests required to test all our hypotheses. Our plan is to estimate a single model to test each hypothesis, and sometimes estimate one model to test two hypotheses (H3 and H4). Thus, we do not think we run into multiple comparisons issues, since the multiple testing problem applies to families of tests that test the same hypothesis/theory, not to all tests in a paper. The only solution to limit the number of tests here would be to reduce the number of hypotheses we test. In the most extreme case, we could acquire a new data set to test each hypothesis, which would minimize the number of tests per study, but would seem rather wasteful and not very desirable. Thus, we think it is best to test as many hypotheses as make sense for a given dataset, and correct for multiple comparisons within each hypothesis.

Regarding model complexity, e.g. the number of factors entered into a model, we think that our models can indeed be made somewhat simpler. At the moment, we enter time (pre, post) as a factor in most of our ANCOVAs. We changed the analysis approach throughout the manuscript, so that instead of using time as a factor, we compute a difference score (e.g. $FWB_{post} - FWB_{pre}$) and use these difference scores as dependent variables. This simplifies our statistical models considerably. Although we changed all analyses that include time as a factor, here we describe our approach in more detail for Hypotheses 3 and 4:

In these hypotheses we want to test moderation effects, which necessitates one main effect (group) and at least one potential moderator (e.g. baseline attitudes). Originally, we planned to test Hypotheses 3 and 4 with the following ANCOVA: $FWB \sim \text{time (pre,post)} * \text{group (AntiFW, Control)} * FWB_{pre} * FWA_{pre}$. A time * group * FWB_{pre} interaction effect would demonstrate that manipulation effect strength depends on FWB_{pre} (Hypothesis 3), and Hypothesis 4 can be tested using a similar logic. Interpreting 3-way interactions is not always straightforward, and computing difference scores simplifies the model considerably here. Using these difference scores ($FWB_{post} - FWB_{pre}$) as dependent measures, we can assess the manipulation with a simple main effect of group (AntiFW, Control), instead of a 2-way interaction (group * time). This allows us to simplify the ANCOVA to these factors: $\text{group (AntiFW, Control)} * FWB_{pre} * FWA_{pre}$. In order to show that e.g. FWA_{pre} affects the manipulation, we only need to show a group * FWA_{pre} interaction, which is more easily interpretable and more intuitive than in our original analysis.

Reviewer 3

In this proposed study, the authors seek a better understanding of lay views of free will by considering both attitudes toward free will and free will beliefs. Past work in this area has exclusively focused on free will beliefs, leaving open the possibility that experimental manipulations have different effects for attitudes than for beliefs or that the degree to which someone values the idea of free will moderates the effect of manipulations on free will beliefs.

The proposed data collection and analysis plan are feasible and sound. The study design uses established measures and manipulations. This research promises to yield valuable insights and help clarify unsettled issues in the literature (cf. Buttrick et al., 2020). While the focus on US adults is

consistent with prior work in this area, I suggest adding some discussion and consideration of research showing considerable cross-national variability in the strength and correlates of free will beliefs.

We thank the reviewer for the positive evaluation of our work. We fully agree that cross-national variability is substantial in free will beliefs, in fact we published on this issue in the past (<https://journals.plos.org/plosone/article?id=10.1371/journal.pone.0221617>). We will discuss our results in light of expected cultural differences in the stage 2 registered report.

My only criticism of the proposal is that the proposed hypotheses are left rather vague. I suggest revisions to the introduction to clarify links to theory, extant research, and the three research questions for the five hypotheses listed in the analysis plan (e.g., reframe and expand the discussion on the first full paragraphs of p. 5 and p. 6 to relate to the specific hypotheses to be tested).

We revised the complete introduction section to now explicitly link it to the hypotheses listed in Table 1. This should clarify how the points we raise in the introduction are linked to the specific hypotheses we formulated. We also added additional citations to more clearly link the current investigation to existing theory and empirical work.

Dr. David Wisniewski
E david.wisniewski@ugent.be
Henri Dunantlaan 2
9000 Gent
Belgium
www.ugent.be

DATE

15 November 2021

Dear Editor,

Please find enclosed our manuscript titled 'Relating free will beliefs and attitudes' by Wisniewski, Cracco, González-García, and Brass, which we re-submit for consideration as a Stage 2 Registered Report to the *Royal Society Open Science*.

We now carried out the research, as pre-registered in the Stage 1 Registered Report which was granted in principle acceptance. Our findings confirmed some key hypotheses, showing that free will beliefs and attitudes are closely related. Results also showed some unexpected patterns however, showing that free will attitudes remain unaffected by a free will belief manipulation, and that baseline beliefs and attitudes had negligible effects on the strength of the manipulation.

We would like to point out that we detected a small error in the approved protocol while carrying out the research. The exploratory analysis 2, which tests for potential effects of demographic variables on free will beliefs and attitudes, was not designed correctly and some potential effects were left out of this analysis. On page 15 of the manuscripts, we transparently point this out to the reviewers, and we adapted the analysis to now include tests for all demographic variables. Of course, we are open to alternative suggestions by either the editor or reviewers.

Overall, we believe this paper is a significant contribution to the literature on free will beliefs, and look forward to your feedback.

With kind regards,

David Wisniewski, Emiel Cracco, Carlos González-García, Marcel Brass

Appendix D

Thank you for giving me the possibility to review this article. I was quite interested in seeing the result in the first round, and now I feel even more interested by what we can do with it. In this sense, I find the discussion a little fast and superficial. By this I mean that it sticks very much to the results and does not offer a more general interpretation of the link between beliefs and attitudes, but also of how to think about attitudes related to free will. I propose two ways to improve the discussion below. I also found several (minor) ways to improve the paper that I detail in general and specific sections.

General:

I felt a bit confused by the use of “se” through the whole manuscript. Indeed, sd are not presented in the manuscript and it is a problem when researchers will want to use this study in meta-analysis or systematic review. If think you can replace se with sd, or, as I propose in the result section, provide a table of descriptive and a summary of findings.

For the whole manuscript (result and discussion sections), you provide cohen’s d and pearson’s r coefficient without confidence intervals. Please add them because an effect size alone is meaningless and need to be put into perspective with the range of the estimate. It is easy to implement because you use the cohen.d package. When you compute the effect size (your variable ddiff), you have a parameter called conf.int with 2 possibilities, [1] is 5% and [2] is 95%.

So for example you can replace you code lines 493 to 502 with the following (copy paste):

```
tab = map_df(list(moddiff), tidy)
```

```
tab$test = c('t.paired')
```

```
tab$d = c(ddiff$estimate)
```

```
tab$d5=c(ddiff$conf.int[1])
```

```
tab$d95=c(ddiff$conf.int[2])
```

```
tab = tab[c('test','statistic','parameter','p.value','d','d5','d95')]
```

```
names(tab)=c('test','t.value','df','p.value','d.value','IC5%','IC95%')
```

```
kbl(tab,caption = 'Manipulation check: FWB - post group comparison', booktabs = T,escape  
=F, digits=2) %>%
```

```
kable_classic(full_width = F, html_font = "Cambria")
```

In your table, you will find the 95% confidence interval for cohen's d. It is really important to add it for reproducibility power analysis and synthesis of evidence.

Specific:

- Intro

- The end of page 3 has a structure problem; I think the sentence is broken. There is also one parenthesis without beginning.
- The beginning of page 3, where it is said that “recent neuroscientific findings apparently demonstrate that our choices are determined by unconscious brain activity” Would it be possible to discuss it regarding the Brass, Furstenberg and Mele work (2019) work not cited in this paper ?

- Results

- You can improve significantly the result section by adding a descriptive table for 1 : overall looking at the results and 2 : making appropriate power-analysis or meta-analysis. You already have it line 163, 192 and 220 of the R code. Please add it. It will also answer to the problem of providing only the standard error in-line.
- Page 16 it is said that FWB scores were very high, FWA score were even higher. I remarked at this moment that FWA had a lower SD than FWB. This is something you should mention since it seems that people are very “polarized” in their attitude, not that much in their beliefs.

- page 17 in the non-pre-registered analyses, line 2 there are two “will”
- Page 17 you said you used cocor to compare correlations. You made a table for the cocor differences in ligne 342, please share it in the manuscript.
- At the end of page 17, I read “since the lower variance might limit correlations”, this should be more discussed: there is a curious difference between FWA and FWB as data are more “normally distributed” for FWB. Add a possibility of explanation in discussion. This is the same remark that when I said the SDs are different.
- Page 23, I see that you provided the Cronbach’s alpha. More and more studies are showing that McDonald’s omega is more precise; please add it (see Deng, L., & Chan, W. (2017). Testing the difference between reliability coefficients alpha and omega. *Educational and psychological measurement*, 77(2), 185-203.)

- Discussion

- I would like a paragraph explaining the limitation of the operationalization of Attitude, given that it is operationalized as hypothetical attitudes. What is the difference with other operationalization of attitude in psychology or in experimental philosophy? What would be the limitations of operationalize an attitude in an hypothetical way etc...
- I also would appreciate a bit about the “overserved ceiling effect” of attitude: why is that happening and why beliefs data seems more “normal” with a higher SD.
- About the religious belief, I wanted to point out that it is from an American point of view. In France, we would definitely have a clear difference from Christian people since most French people are not practicing Christians. Thus, it is also a cultural variable and not only a religion variable.

- Reference

- Page 32 at the beginning the what’s with free will is in bold

- Idem page 33 meta-analysis on belief in free will manipulations
- Idem page 36 different types of religiosity

Finally, the three main points of revisions are the need for SDs, either in text or in a descriptive table, the confidence intervals for effect sizes, and a discussion about what you exactly measured with your measure of attitude and how it can be improved in future studies. Since you already have these data and you can add it fast, I recommend minor revision.

Appendix E

Dear Editor,

Thank you for the positive appraisal of our work, and for assessing any final, minor changes at desk. This will speed up the review process significantly. Please find our point-by-point responses to Reviewer 2 below. We included a “tracked changes” version of our manuscript, where we marked all changes for your convenience.

With kind regards,

David Wisniewski

Reviewer 2:

Thank you for giving me the possibility to review this article. I was quite interested in seeing the result in the first round, and now I feel even more interested by what we can do with it. In this sense, I find the discussion a little fast and superficial. By this I mean that it sticks very much to the results and does not offer a more general interpretation of the link between beliefs and attitudes, but also of how to think about attitudes related to free will. I propose two ways to improve the discussion below. I also found several (minor) ways to improve the paper that I detail in general and specific sections.

Thank you for the thorough and constructive review, we appreciate your comments and provide point-by-point responses below.

General:

I felt a bit confused by the use of “se” through the whole manuscript. Indeed, sd are not presented in the manuscript and it is a problem when researchers will want to use this study in metaanalysis or systematic review. If think you can replace se with sd, or, as I propose in the result section, provide a table of descriptive and a summary of findings.

We agree with this point, and replaced SE with SD throughout the whole manuscript.

For the whole manuscript (result and discussion sections), you provide cohen’s d and pearson’s r coefficient without confidence intervals. Please add them because an effect size alone is meaningless and need to be put into perspective with the range of the estimate. It is easy to implement because you use the cohen.d package. When you compute the effect size (your variable ddiff), you have a parameter called conf.int with 2 possibilities, [1] is 5% and [2] is 95%.

So for example you can replace you code lines 493 to 502 with the following (copy paste):

```
tab = map_df(list(moddiff), tidy)
```

```
tab$test = c('t.paired')
```

```
tab$d = c(ddiff$estimate)
```

```
tab$d5=c(ddiff$conf.int[1])
```

```
tab$d95=c(ddiff$conf.int[2])
```

```
tab = tab[c('test','statistic','parameter','p.value','d','d5','d95')]
names(tab)=c('test','t.value','df','p.value','d.value','IC5%','IC95%')
kbl(tab,caption = 'Manipulation check: FWB - post group comparison', booktabs = T,escape
=F, digits=2) %>%
kable_classic(full_width = F, html_font = "Cambria")
```

In your table, you will find the 95% confidence interval for cohen’s d. It is really important to add it for reproducibility power analysis and synthesis of evidence.

Thank you for pointing this out, and for providing helpful code suggestions. We integrated the code and now provide 95% confidence intervals for all Cohen’s ds and correlation coefficients reported in the manuscript.

Specific:

- Intro

o The end of page 3 has a structure problem; I think the sentence is broken. There is also one parenthesis without beginning.

We fixed the issue with the parentheses and the broken sentence.

o The beginning of page 3, where it is said that “recent neuroscientific findings apparently demonstrate that our choices are determined by unconscious brain activity” Would it be possible to discuss it regarding the Brass, Furstenberg and Mele work (2019) work not cited in this paper ?

Thank you for pointing out this fitting additional citation. Indeed, it would be interesting go into a bit more detail on what neuroscience can and cannot tell us about free will. Unfortunately, since the introduction section cannot be altered in stage 2 of a registered report (beyond e.g. fixing typos), we are unable to implement such changes to the text at this time.

- Results

o You can improve significantly the result section by adding a descriptive table for 1: overall looking at the results and 2: making appropriate power-analysis or metaanalysis. You already have it line 163, 192 and 220 of the R code. Please add it. It will also answer to the problem of providing only the standard error in-line.

We agree, and now added a results table on page 17:

	Pre session				Post session - Control group				Post session - AntiFW group			
	mean	median	sd	se	mean	median	sd	se	mean	median	sd	se
FWB	0.28	0.33	0.45	0.01	0.25	0.33	0.46	0.02	0.18	0.2	0.49	0.02
FWA	0.65	0.67	0.31	0.01	0.68	0.73	0.3	0.01	0.65	0.73	0.33	0.01
DUB	0.19	0.27	0.52	0.02	0.27	0.4	0.53	0.02	0.24	0.33	0.53	0.02
DUA	0.31	0.33	0.43	0.01	0.4	0.47	0.43	0.02	0.42	0.47	0.42	0.02
DEB	-0.39	-0.3	0.44	0.01	-0.26	-0.27	0.45	0.02	-0.25	-0.27	0.45	0.02
DEA	-0.32	-0.33	0.43	0.01	-0.33	-0.33	0.46	0.02	-0.34	-0.33	0.44	0.02

Table 1: Descriptive statistics. The mean, median, standard deviation (sd), and standard error (se) for each belief (FWB, DUB, DEB) and attitude (FWA, DUA, DEA) subscale are shown, separately for the Pre Session, and both groups in the Post Session.

o Page 16 it is said that FWB scores were very high, FWA score were even higher. I remarked at this moment that FWA had a lower SD than FWB. This is something you should mention since it seems that people are very “polarized” in their attitude, not that much in their beliefs.

We added an extra sentence on page 16, explicitly pointing out the numerical difference in variances between FWBs and FWAs:

At the same time, the variance in FWA scores was somewhat lower numerically than the variance in FWB scores.

o page 17 in the non-pre-registered analyses, line 2 there are two “will”

We fixed this typo, and removed the additional ‘will’.

o Page 17 you said you used cocor to compare correlations. You made a table for the cocor differences in ligne 342, please share it in the manuscript.

We added an additional table on page 19, reporting these results:

Comparison	difference	z	p
FW - DU	0.35	12.34	<0.001
FW - DE	0.4	13.82	<0.001
DU - DE	0.04	2.2	0.03

Table 2: Comparison of belief-attitude correlations. We computed the belief-attitude correlation for each subscale, FW = r(FWB,FWA), DU = r(DUB, DUA), DE = r(DEB, DEA), and then computed difference scores between subscales (difference). These differences were assessed statistically using *cocor::cocor*, and both z-values and p-values are reported.

o At the end of page 17, I read “since the lower variance might limit correlations”, this should be more discussed: there is a curious difference between FWA and FWB as data are more “normally distributed” for FWB. Add a possibility of explanation in discussion. This is the same remark that when I said the SDs are different.

We agree that this difference in sample variance is curious and deserves some explanation. In our manuscript, we already provide a tentative explanation on page 28 (discussion section), where we speculate that this might be explained with differences in confidence:

One potential explanation for this difference in variance might instead be that participants were more confident of their free will attitudes than they were of their free will beliefs, leading to more consistent responses. Given the lack of a direct measure of confidence or certainty, this remains speculative at the moment, but future research should directly address this issue by adding within-subjects measures of attitude/belief certainty, instead of merely relying on sample variance.

o Page 23, I see that you provided the Cronbach’s alpha. More and more studies are showing that McDonald’s omega is more precise; please add it (see Deng, L., & Chan, W. (2017). Testing the difference between reliability coefficients alpha and omega. *Educational and psychological measurement, 77(2)*, 185-203.)

We now amended Supplementary Table 1 to also include McDonald’s omega, and reference this in the manuscript explicitly on page 24.

Supplementary Table 1: Internal consistency.

	Belief		Attitude	
	α	ω	α	ω
Free Will	0.85	0.87	0.79	0.82
Determinism	0.8	0.83	0.77	0.82
Dualism	0.85	0.88	0.78	0.82

Cronbach’s alpha (α) and McDonald’s omega (ω) for each subscale of belief and attitude version of the FWI (Nadelhoffer et al. 2014). We only used data from the first time point to compute these values.

Page 24:

Using McDonald’s omega led to comparable, slightly higher estimates of internal consistency, $\omega = [0.82 - 0.88]$ (Supplementary Table 1).

- Discussion

o I would like a paragraph explaining the limitation of the operationalization of Attitude, given that it is operationalized as hypothetical attitudes. What is the difference with other operationalization of attitude in psychology or in experimental philosophy? What would be the limitations of operationalize an attitude in an hypothetical way etc...

We would like to point the reviewer to the Methods section, where we motivate the methodological approach, specifically talking about the use of hypothetical items and how this differs from previous research. Following this comment, we now made the limitations of our approach and its differences to previous research even more explicit.

Page 10

We would like to point out that all attitude items were designed as hypothetical statements (e.g. “I would like if people always had the ability to do otherwise.”). While this did make the items somewhat longer and challenging, we chose this approach to ensure that attitude items did not assume either a belief or a disbelief in the item in question. While “I like that people always have the ability to do otherwise.” can be a good measure of FWAs in principle, it assumes that people believe free will exists, which might be a problem especially for participants who do not believe so. This problem is avoided when using hypothetical statements, which we chose to do here.

Page 11

One key advantage of using this procedure is that both beliefs and attitudes are measured using similar methods, making measures highly comparable. In a previous study, FWAs were measured using a somewhat different approach (Cracco et al., 2020). Participants were asked whether they felt positively/negatively, and warm/cold about free will, and whether they liked it or not. This approach is widely used in attitude research since it is simple and intuitive (Nosek, 2007). Yet, while asking participants whether they like free will is indeed somewhat intuitive, that is not the case for determinism and dualism. Many people might not know about the concept of dualism at all and thus could not assess whether or not they like it. For this reason, we decided to deviate from this approach. Instead of directly asking about attitudes towards the underlying constructs (free will, determinism, dualism), we followed the Free Will Inventory and asked about attitudes towards statements associated with these constructs, assuming that asking participants e.g. “I would like to live in a world where people always have free will” would be more intuitive and preferable.

o I also would appreciate a bit about the “overserved ceiling effect” of attitude: why is that happening and why beliefs data seems more “normal” with a higher SD.

This point seems similar to the point raised above in the Results section, please see our response there.

o About the religious belief, I wanted to point out that it is from an American point of view. In France, we would definitely have a clear difference from Christian people since most French people are not practicing Christians. Thus, it is also a cultural variable and not only a religion variable.

We agree that results reported here should be seen as specific to the USA, and cannot be easily generalized to other cultures. We now state this fact more explicitly in the manuscript on page 31:

Additionally, since we only investigated participants from the USA, we cannot rule out that differences between religious and non-religious participants might be specific to this culture. Results might differ in e.g. European or Asian samples.

- Reference

o Page 32 at the beginning the what’s with free will is in bold

o Idem page 33 meta-analysis on belief in free will manipulations

o Idem page 36 different types of religiosity

We fixed the issues in the reference section.

Finally, the three main points of revisions are the need for SDs, either in text or in a descriptive table, the confidence intervals for effect sizes, and a discussion about what you exactly measured with your measure of attitude and how it can be improved in future studies. Since you already have these data and you can add it fast, I recommend minor revision.